# Reasoning with Preference Constraints: A Benchmark for Language Models in Many-to-One Matching Markets

**Marylou Fauchard**                                    *marylou.fauchard@mila.quebec*
*Mila, Université de Montréal*

**Florian Carichon**                                    *florian.carichon@mila.quebec*
*Mila, McGill University*

**Margarida Carvalho**                                  *carvalho@iro.umontreal.ca*
*Mila, Université de Montréal*

**Golnoosh Farnadi**                                    *farnadig@mila.quebec*
*Mila, McGill University, Université de Montréal*

**Reviewed on OpenReview:** *https://openreview.net/forum?id=2dpt2Ughzt*

## Abstract

Recent advances in reasoning with large language models (LLMs) have demonstrated strong performance on complex mathematical tasks. Techniques such as Chain-of-Thought and In-Context Learning have further enhanced this capability, making LLMs both powerful and accessible tools for a wide range of users, including non-experts. However, their application to problems arising in operations research, particularly those at the intersection of combinatorial optimization and game theory that require domain expertise, remains underexplored. To address this gap, we introduce a benchmark of 369 instances for the College Admission Problem, a canonical many-to-one matching problem that requires reasoning about participants' preferences, stability, feasibility, and optimality. We evaluate several LLMs, both reasoning-specialized and more traditional, defined here as models used without any dedicated reasoning mechanisms. Even though no prompt consistently offered the best performance, using strategies such as Chain-of-Thought, In-Context Learning and role-based prompting, reasoning LLMs reacted differently from the traditional ones. While reasoning-enhanced models generally outperform traditional ones, most still exhibit performance degradation as problem complexity increases, whereas traditional models show a more fundamental lack of understanding of key concepts such as stability. Finally, we report the performance from iterative prompting with auto-generated feedback and show that they are not always monotonic; they can peak early and then significantly decline in later attempts. Overall, this work offers a new perspective on model reasoning performance and the effectiveness of prompting strategies in combinatorial optimization problems with preferential constraints.

## 1 Introduction

Recent advancements in large language models (LLMs) have revealed emergent reasoning capabilities, enabling them to address increasingly complex problems. Most notably, LLMs have been applied to solve mathematical problems that require step-by-step analytical reasoning (Ahn et al., 2024). They have also been used in complex game-theoretic scenarios with interactions and strategic decision-making (Sun et al., 2025a) and combinatorial optimization tasks (Yang et al., 2023; Wang et al., 2023a), proving that LLMs can extend their reasoning abilities to various kind of complex real-world scenarios. To rigorously assess the effectiveness of LLM reasoning and its associated methods, a range of benchmarks and evaluation metrics have been developed. These benchmarks can be categorized into (1) objective, ground-truth-based benchmarks such as

MATH (Hendrycks et al., 2021) or SCIBENCH (Wang et al., 2023b) and (2) subjective, preference-based evaluations such as Chatbot Arena (Chiang et al., 2024). Although these benchmarks are valuable for understanding LLMs' capacity to solve these theoretical problems, they are still limited due to static question sets, the risk of data contamination, a lack of real-world complexity, high variance, or prioritizing form over reasoning quality (Chen et al., 2025a). Combinatorial optimization benchmarks such as HeuriGym (Chen et al., 2025a), CO-Bench (Sun et al., 2025b), or FrontierCO (Feng et al., 2025) provide ideal evaluation settings as they involve multi-step and iterative reasoning over the large solution spaces of NP-hard problems, resist memorization well, and offer robust quantitative metrics (Sartori & Blum, 2025; Chen et al., 2025a). While these benchmarks show that LLMs hold promise in solving NP-hard problems with cost-minimization objectives, they overlook a critical class of objective combinatorial problems involving preference-based constraints, typical of matching problems where stability among multiple participants is the primary measure of success.

Two-sided many-to-one matching problems, which are solvable in polynomial time and involve satisfying preferences and capacity constraints, provide a natural benchmark for testing LLMs' reasoning abilities. First, non-experts often address this optimization problem in several contexts, such as College Admission (Sartori & Blum, 2025), Hospital-Resident matching (Roth, 1984), or School Choice (Sönmez et al., 2003). These problems' model has been widely adopted by institutions to represent participants' preferences, such as in the US (Abdulkadiroğlu et al., 2005a;b; Roth, 1984), Hungary (Biró, 2008), and Germany (Braun et al., 2007). Secondly, the problem involves providing a matching, formally defined in Section 3.1, that satisfies desirable properties regarding the students' and colleges' preferences, introducing an additional layer of reasoning complexity. Mathematically, the optimal solution corresponds to finding a matching that minimizes the sum of ranks for students, such that no college exceeds its capacity and that no student-college pair prefers to be matched to each other over their assigned outcomes. Such a solution is considered student-optimal, feasible, and stable (Gale & Shapley, 1962). This paper investigates how LLMs represent and reason about preferences and constraints in matching problems to produce stable, student-optimal outcomes. In the remainder of this article, we will specifically address the College Admission problem (CAP). Our contributions are the following:

- We introduce a controlled dataset of CAP instances involving participants, comprising both students or colleges, with their associated preferences and capacities.

- We propose four objective metrics specifically designed for evaluating LLMs on matching problems, namely the CAP.

- We empirically evaluate several LLMs from the LLaMA, Qwen, Mistral, and GPT families across multiple prompting strategies, assessing their performance when used as a centralized coordinator for many-to-one matching problems.

- We rigorously analyze how instance-level factors, such as the number of students, preference completeness, and capacity regimes, affect the satisfaction of formal properties, revealing where selected models fail to provide matchings or simulate the associated algorithm.

- We investigate how prompting approaches, such as Chain-of-Thought, In-Context Learning, and iterative prompting, influence the reasoning of LLMs in solving matching problems.

## 2 Related Work

This work builds on two key areas of related research: two-sided many-to-one matching markets and recent developments and benchmarks for LLMs' reasoning. We defer the discussion of related work on reasoning over preferences in agent-based systems to Appendix A.

### 2.1 Two-Sided Many-to-One Matching Markets

Matching markets are mechanisms designed to pair participants on two sides of a market based on their preferences. A foundational model is the one-to-one matching problem, famously formalized by Gale and

Shapley in the context of the Stable Marriage Problem (Gale & Shapley, 1962), where each participant on one side is matched with exactly one on the other. Many real-world applications, however, fall under the many-to-one setting. This setting mainly allows an unequal number of participants on both sides, with different capacities making the problem more general. The Deferred Acceptance (DA) algorithm, introduced by Gale and Shapley (Gale & Shapley, 1962), has become a foundational tool for solving both one-to-one and, its generalization, many-to-one matching problems. The resulting matching is guaranteed to be feasible, stable, and student-optimal, notions that will be formally defined in Section 3.1. This algorithm has been implemented in several real-world settings (Abdulkadiroğlu et al., 2005a;b; Roth, 1984; Biró, 2008). They demonstrate that a diversity of institutional constraints, such as stability or equitable access, exists in the College Admission problem, and more generally in many-to-one problems. Recent work has focused on extensions and generalizations of these matching problems by introducing additional layers of complexity, such as allowing ties in preferences and enforcing common quotas (Ágoston et al., 2022), considering flexible capacities (Bobbio et al., 2026; 2025; Afacan et al., 2024) or contingent priorities (Rios et al., 2024). While existing datasets, such as the Chilean school choice dataset (Correa et al., 2019), provide valuable real-world data, they do not offer sufficient control over problem instance parameters to effectively evaluate an LLM's reasoning and understanding.

## 2.2 LLMs Reasoning Benchmarks

LLMs have recently demonstrated strong capabilities to tackle complex reasoning tasks in various fields (Parashar et al., 2025). In the context of mathematical reasoning, LLMs are primarily evaluated for their capabilities in logical analysis, deductive reasoning, and arithmetic operations (Yuan et al., 2023). Standard benchmarks such as MATH (Hendrycks et al., 2021), Olympiad-Bench (He et al., 2024), or MathBench (Liu et al., 2024) have become central in evaluating reasoning capabilities of LLMs. Models have achieved high performances on traditional benchmarks, raising concerns about memorization instead of true reasoning capabilities with multi-step deduction, calling for a new perturbed dataset (Huang et al., 2025). Alongside these benchmarks, combinatorial optimization problems offer a rich ground for evaluating complex reasoning. They exhibit specific characteristics: they often have numerous potential solutions that are not always directly comparable, involve different constraints on variables, and the optimization process can encounter multiple local optima (Sartori & Blum, 2025). Within this framework, LLMs have been tested for heuristic generation (Ye et al., 2024), algorithm design (Sartori & Blum, 2025), re-optimization (Ye et al., 2026), and direct problem-solving via prompt-based techniques (Guo et al., 2023; Yang et al., 2023). The first existing benchmarks focus on the classical and potentially overused Traveling Salesman Problem (Sartori & Blum, 2025; Ye et al., 2024; Yang et al., 2023). Other benchmarks mainly assess reasoning through real-world scenarios based on graph and set optimization problems (Wang et al., 2023a; Feng et al., 2025), pairing and scheduling problems (Chen et al., 2025a; Sun et al., 2025b), or machine learning related problems such as logistic regression or grid search optimization (Guo et al., 2023; Yang et al., 2023). Most of these benchmarks predominantly target well-known NP-hard combinatorial problems centered on cost minimization heuristics. Therefore, they neglect structured tasks, such as problems with preference-based constraints and stability requirements. A first step toward addressing constraint-driven problems has been explored with the Transportation and Assignment Problems (Khan & Hamad, 2024). However, although the underlying mathematical formulations support many-to-one matchings, the prompt design appears to restrict the problem to one-to-one instances. Recent work has also studied LLMs understanding of one-on-one matching market with rigorous evaluation (Hosseini et al., 2025). Yet many-to-one market offers a richer testbed for evaluating LLM reasoning, due to the presence of potentially unmatched participant and more complex capacity constraints. By comparing several prompting strategies, our benchmark exposes the limitations of several open-weight LLMs in tackling complex combinatorial optimization tasks but also reveals interesting dynamics for evaluating reasoning complexity and model understanding in structured decision-making contexts.

# 3 Methodology

## 3.1 Problem Statement

In the College Admission problem, we have a set of students $\mathcal{S}$ and a set of colleges $\mathcal{C}$. Each student in $s \in \mathcal{S}$ has a strict preference order $\succ_s$ over $\mathcal{C} \cup \{\emptyset\}$. We write $c_i \succ_s c_j$ for $c_i, c_j \in \mathcal{C}$ if student $s$ prefers college $c_i$ over $c_j$. If student $s$ prefers to be unassigned than being matched with a college $c \in \mathcal{C}$, we write $\emptyset \succ_s c$. Therefore, the student's preference list results in a ranking of the colleges. We denote by $rank_s(c)$ the rank of a college $c \in \mathcal{C}$ in the preference list of student $s$, e.g., the most preferred college for student $s$ has rank 1. Similarly, each college $c \in \mathcal{C}$ has a strict order $\succ_c$ over $\mathcal{S} \cup \{\emptyset\}$. Moreover, a college $c$ has a capacity $q_c \in \mathbb{Z}_+$. We denote $\mathcal{E} \subseteq \mathcal{S} \times (\mathcal{C} \cup \{\emptyset\})$ as the set of acceptable pairs, meaning that a pair is not acceptable, i.e $(s,c) \notin \mathcal{E}$, if $\emptyset \succ_s c$ or $\emptyset \succ_c s$.

An assignment is any subset $\mathcal{M} \subseteq \mathcal{E}$, interpreted as any set of student–college pairs. An assignment does not necessarily satisfy capacity constraints. An assignment $\mathcal{M}$ is *assignment stable* if for no pair $(s,c) \in \mathcal{E} \setminus \mathcal{M}$, (i) student $s$ prefers $c$ over their current matching, i.e., $c \succ_s \mathcal{M}(s)$ where $\mathcal{M}(s)$ is either the school assigned to student $s$ or $\emptyset$ when the student is unmatched, and (ii) school $c$ is under-subscribed, i.e., $|\mathcal{M}(c)| < q_c$ where $\mathcal{M}(c)$ is the set of students assigned to $c$, or prefers student $s$ over some student $s' \in \mathcal{M}(c)$, i.e., $s \succ_c s'$. A pair that would respect either of the previous conditions is called a blocking pair and prevents stability.

A set $\mu \subseteq \mathcal{E}$ is called a *feasible matching* when (i) each student $s \in \mathcal{S}$ is matched to at most one school, i.e., $|\{c \in \mathcal{C} : (s,c) \in \mu\}| \leq 1$, and (ii) each school $c \in \mathcal{C}$ is matched to no more students than its capacity, i.e., $|\mu(c) = \{s \in \mathcal{S} : (s,c) \in \mu\}| \leq q_c$. For clarity, we will refer to feasible matchings as "matchings" in the remainder of the paper. A matching $\mu$ is *matching stable* if for no pair $(s,c) \in \mathcal{E} \setminus \mu$, (i) student $s$ prefers $c$ over their current matching, i.e., $c \succ_s \mu(s)$ where $\mu(s)$ is either the school assigned to student $s$ or $\emptyset$ when the student is unmatched, and (ii) school $c$ is under-subscribed, i.e., $|\mu(c)| < q_c$ where $\mu(c)$ is the set of students assigned to $c$, or prefers student $s$ over some student $s' \in \mu(c)$, i.e., $s \succ_c s'$. By default, throughout the paper, the term stability will refer to matching stability. Matching stability requires both feasibility and assignment stability. In contrast, assignment stability does not require feasibility and is therefore a weaker notion. Finally, a stable matching $\mu$ is *student-optimal* when no student can get a better assignment in any other stable matching. It is known that a student-optimal stable matching $\mu$ coincides with the stable matching minimizing the sum of the ranks of the students, i.e., $\sum_{(s,c) \in \mu} rank_s(c)$. With these formalizations, we define our College Admission problem as computing the student-optimal stable matching.

## 3.2 Benchmark

We introduce a new benchmark dataset explicitly designed for controlled experimentation. Compared to the Chilean dataset (Correa et al., 2019), which focuses on large-scale problems, our benchmark provides a controlled framework over instance parameters, enabling systematic evaluation on well-defined problem settings. This controlled environment enables detailed analysis of model reasoning and decision-making processes, which would be infeasible for larger instances that exceed the context window of current LLMs. Additionally, the code to generate the benchmarks data[1] allows users to scale instance complexity by adjusting the number of students, colleges, and other parameters to accommodate broader generalization experiments. Our benchmark is designed to be scalable, allowing easy extension to real-world sizes while remaining compatible with other experimental parameters. However, we note that scalability is not the primary concern on most models since, as the experiments will reveal, they struggle to produce student-optimal matchings even on small instances. Nevertheless, the benchmark is also intended to identify models that successfully capture the core concepts underlying matching problems on these controlled instances. Such understanding on controlled small-scale instances provides evidence that motivates investigating these models on larger-scale matching problems and, more broadly to complex combinatorial optimization tasks, such as heuristic generation for NP-hard problems. Finally, providing the code to generate new instances is a key measure to mitigate data contamination, a prevalent issue in benchmark evaluation, by ensuring models are tested on previously unseen instances.

---

[1]The link to the GitHub is: https://github.com/maryloufauchard/CAP_Benchmark

In this article, we generate synthetic instances of the CAP by varying the number of students, the number of colleges represented through the student–college ratio, the preference structure of students, and the total capacity of the colleges. More specifically, we consider the following parameters:

- Number of students: 5, 10, 15, 20
- Student-College ratio: 1:1, 2:1, 3:1, 4:1
- Preference type: Incomplete, Flexible, Complete
- Total capacity: Under-capacity, Exact-capacity or Over-capacity

Each configuration is tested under three types of student preference structures:

- **Complete Preferences** where each student ranks all available colleges;
- **Incomplete Preferences** where each student ranks a fixed number of colleges, strictly fewer than the total available; and
- **Flexible Preferences** where the length of each student's preference list is drawn randomly between a predefined minimum and the total number of colleges.

Since college preferences do not impact the complexity of the DA algorithm, they will always remain complete, following the Chilean dataset structure. Therefore, the preference of each college will be drawn from a uniform random permutation of all students; this is realistic in school choice applications using random tiebreaker (Correa et al., 2019). Similarly for each student, we draw a uniform random permutation of the colleges and truncate it at the specified preference length. While real-world scenarios may have correlated preferences, our goal is to control the difficulty of the instances for evaluating LLMs under the standard DA algorithm's complexity. The latter runs in $O(S \cdot P)$, where $S$ is the number of students and $P$ is the maximal student preference length. In the extreme case where all students share the same preference list, the complexity becomes linear in $P$. Empirically, we observe a significantly larger number of DA iterations for instances with preferences generated via a uniform distribution than for correlated preferences generated via the Mallows distribution (Tang, 2019). Therefore, using independent uniform preferences avoids bias or confounding from additional structural parameters, allowing us to focus solely on how the LLMs reason over preferences, relative to the DA algorithm. In this paper, the total capacity varied to assess whether the LLM is able to leave some students unassigned when necessary, and to examine whether capacity acts as a confounding factor in the model's decision-making. It will therefore be tested across three settings:

- **Under-capacity** with a capacity of 80% of the number of students, forcing some students to be unassigned;
- **Exact-capacity** where the total capacity equals the number of students, and
- **Over-capacity** where total capacity exceeds the number of seats by a fixed amount of 10.

We test this last configuration since adding capacity to colleges having already empty seats after all students are assigned should not change the solution, assuming that preferences remain unchanged, if the model reasons correctly. Under all settings, colleges are assigned random capacities, ensuring that each college has at least one available seat. We also explore different student-college ratios (1:1, 2:1, 3:1, 4:1), with some configurations omitted when the total capacity falls below the number of colleges. For each combination of parameters, we generate three random seeds while creating the preferences to account for variability. In total, our benchmark comprises 369 instances, with some configurations omitted due to infeasibility. A more detailed breakdown of the number of instances is included in Appendix B.

Figure 1 presents an example instance, illustrating the exact input format under the following parameter settings: 10 students, a student–college ratio of 4:1, complete preferences, and an under-capacity setting with a total of 8 available seats. This configuration yields 3 colleges, computed as 10/4 rounded up. All other instances are constructed and presented analogously for different parameter configurations. Figure 2 presents the student-optimal matching given by the DA algorithm under this instance.

```
# Num. students:10
# Num. colleges:3
# Students:s1,s2,s3,s4,s5,s6,s7,s8,s9,s10
# Colleges:c1,c2,c3
# Capacities:
c1 3
c2 2
c3 3
# Student preferences:
s1 (1,c3) (2,c2) (3,c1)
s2 (1,c1) (2,c2) (3,c3)
s3 (1,c3) (2,c1) (3,c2)
s4 (1,c1) (2,c3) (3,c2)
s5 (1,c3) (2,c1) (3,c2)
s6 (1,c3) (2,c1) (3,c2)
s7 (1,c3) (2,c1) (3,c2)
s8 (1,c3) (2,c2) (3,c1)
s9 (1,c3) (2,c2) (3,c1)
s10 (1,c2) (2,c3) (3,c1)
# College priorities:
c1 (1,s2) (2,s1) (3,s10) (4,s9) (5,s7) (6,s6) (7,s4) (8,s3) (9,s5) (10,s8)
c2 (1,s9) (2,s8) (3,s6) (4,s5) (5,s10) (6,s1) (7,s4) (8,s3) (9,s2) (10,s7)
c3 (1,s8) (2,s1) (3,s6) (4,s3) (5,s7) (6,s10) (7,s5) (8,s9) (9,s4) (10,s2)
```

Figure 1: Prompt example of a College Admission instance.

```
# DA matching of corresponding input
[("s1", "c3"), ("s2", "c1"), ("s3", "nothing"), ("s4", "nothing"), ("s5", "c2"), ("s6", "c3"), ("s7", "c1"), ("s8", "c3"), ("s9", "c2"), ("s10", "c1")]
```

Figure 2: Example of the solution returned by the DA algorithm.

### 3.3 Prompts

***Prompt strategy*** with distinct formatting significantly influence LLM performance (Mao et al., 2025; Liu et al., 2025). To rigorously evaluate LLM performance on this reasoning task, our 369 instances are augmented with various prompt variations. We evaluate the effect of advanced prompting strategies for LLMs' reasoning, such as Chain-of-Thoughts (CoT) prompting (Chen et al., 2025b), In-Context Learning (ICL) (Dong et al., 2022), and role-based prompting (Sartori & Blum, 2025). More specifically, the prompt containing only the essential information is referred to as the **Basic** prompt. The **General** prompt follows the same guidelines but omits to mention the DA algorithm, testing whether models can succeed without relying on algorithmic instructions. The **Role** prompt follows established prompt templates from prior work (Sartori & Blum, 2025). This component is typically placed at the beginning of the prompt to instruct the model on the intended perspective or behavior (Mao et al., 2025). Figure 3 presents the core structure of the basic prompt, while further details, including the structure of all other prompts, are available in Appendix B.

Two of the prompting strategies fall under the category of ICL. An example is included immediately following the output format component, as commonly adopted in prior work (Mao et al., 2025). The first one, simply named the **ICL**, consists of adding a simplified problem instance along with its corresponding student-optimal matching. To maintain the problem's core structure while adopting a simple example, we use a five-student instance with consistent student-to-school ratios, capacity types, and preference types across different random seeds. The other strategy, referred to as **ICL with steps** in the remainder of the paper, builds upon the

findings of Wei et al. (2022), which suggest that incorporating intermediate reasoning steps into examples can enhance model performance. This strategy is implemented using a fixed example with five students, complete preference lists, and exact capacity settings, where intermediate steps correspond to those of the DA algorithm.

The last four prompting strategies are variations of CoT prompting, which we categorize as **CoT unsupervised**, **CoT text**, **CoT pseudocode**, and **CoT Python**. The CoT unsupervised represents the traditional unguided *Think step-by-step* instruction (Zhang et al., 2025). In contrast, the CoT text variant follows a supervised template that includes an explanation of the solution process (Zhang et al., 2025). The step-by-step explanation is based on a *natural language* description of the DA algorithm introduced in Aziz et al. (2015). Incorporating pseudocode instructions can also improve performance in graph reasoning tasks (Skianis et al., 2024). To investigate whether similar benefits extend to matching problems, we include a pseudocode variation with a description guiding the step-by-step process. Finally, CoT prompting with Python-based, self-describing programs enhances performance on mathematical reasoning tasks (Jie et al., 2023). Accordingly, our CoT Python template includes an implementation of the DA algorithm written in Python.

***Iterative Prompting*** is a multi-step process that iteratively tries to improve LLMs' output by incorporating feedback on the previous failed attempts (Madaan et al., 2023). Prior works have shown promising results from this method that requires no additional training (Ho & Fan, 2025; Yang et al., 2023). More specifically, starting from the original prompt, this technique consists of adding the last model response and feedback to the preceding prompt to improve the following generated answer, up to a predefined maximum of N attempts (Krishna et al., 2024). While much of the work focuses on AI-generated feedback, we use external feedback, which is non-AI-generated, as it is more reliable (Stechly et al., 2023). In our setting, we test iterative prompting, with up to 5 iterations, adding feedback based on whether the previous output verifies our four metrics. Figure 15 illustrates how the feedback is structured to report, for each metric, whether it is satisfied, together with a quantitative indicator of performance to an assessment of the degree to which the criterion is met.

## 4 Empirical Evaluation

We evaluate performance across the following models: (1) **Llama 3 8B** (Dubey et al., 2024) (2) **Llama 3 70B** (Dubey et al., 2024) (3) **Mistral 7B** (Jiang et al., 2023), (4) **Qwen2 7B** (Yang et al., 2024), (5) **QwQ 32B** (Qwen Team, 2025), (6) **GPT-oss 120B** (Agarwal et al., 2025), and (7) **GPT 5 mini** (Singh et al., 2026). The first four models are said to be able to solve complex reasoning problems, whereas the latter have been trained for thinking and reasoning using reinforcement learning and supervised fine-tuning. Therefore, for the remainder of the paper, we will refer to the first four models as the base models and the last three as the reasoning models. We specifically aim to assess their reasoning capabilities in solving the CAP and analyze the impact of model size, model training, and problem complexity on solution quality. Figure 4 introduces our experimental protocol for generating solutions using various LLM models, while details on generation configuration can be found in Appendix D. We selected this range of models for their accessibility and reproducibility, including several open-weight LLMs together with a recent, relatively accessible closed-source model to broaden the comparison. Nevertheless, leveraging on our instance generator, future work should be directed toward evaluating stronger frontier closed-source models and their scaling capabilities, as noted in Section 5.

To evaluate the quality of the generated matchings, we examine four key properties: feasibility, assignment stability, matching stability, and student optimality (recall Section 3.1). Since feasibility only consists of respecting the capacities explicitly mentioned in the instance and assignment stability requires understanding the concept of blocking pairs, implicitly given by the preferences, we consider that the various metrics introduced: (1a) feasibility, (1b) assignment stability, (2) matching stability, and (3) student-optimality form an ascending hierarchy of difficulty (i.e feasibility is the easiest metric while optimality is the hardest one), as each level presupposes the satisfaction of the previous one. A response is considered valid if it adheres to the expected output format of the DA algorithm, allowing for minor acceptable variations. Table 2 presents the

---

**Task**
Implement the student-proposing deferred acceptance algorithm (Gale-Shapley) to solve a many-to-one matching problem for college admissions.

**Instance Format**
The instance will contain the following elements:

- Num. students: Integer, the number of students
- Num. colleges: Integer, the number of colleges
- Students: List of student names
- Colleges: List of college names
- Capacities: Dictionary, each college mapped to its capacity (integer)
- Student preferences: Dictionary, each student is mapped to a list of tuples (rank,college)
- College preferences: Dictionary, each college is mapped to a list of tuples (rank, student)

Ranks are integers where 1 indicates the highest preference.

**Constraints**
The output matching must satisfy the following:
Feasibility:

- Each student is matched to at most one college.
- No college is matched with more students than its capacity.

Stability:

> There must be no blocking pair (student, college) such that:

- The student prefers the college over their current match, and
- The college either has not reached its capacity or prefers the student to at least one of its current matches.

Optimality:

> Among all stable matchings, students are matched to the best possible college based on their preferences.

**Output Format**
Return the matching as a list of tuples of the form (student, match). If a student is unmatched, use "nothing" as the college.

**Instance**
The instance to solve is: [ADD INSTANCE HERE]

**Instruction**
Return only the final matching as a list of tuples.

**Final Matching:**

---

Figure 3: Template of our **Basic** prompt strategy

percentage of valid outputs. Considering only the valid outputs, we compute the proportion of matchings that satisfy each metric. The dataset and the code are available on our GitHub page in footnote 1.

Our evaluation is guided by the following research questions:

**RQ1: To what extent can LLMs generate solutions to many-to-one matching problems that satisfy feasibility, stability, and student-optimality?** Figure 5 presents the proportion of matchings, for each model, that satisfy: feasibility (Figure 5a), assignment stability (Figure 5b), matching stability (Figure 5c), and optimality (Figure 5d) based on the number of students. Table 8 presents the 95% Wilson confidence interval (CI) for each number of students. The results show that the matching problem is a challenging task, where base LLMs struggle even for the most explicit metric. The reasoning LLMs largely outperform the base ones on every metric and across instance sizes, highlighting the need for specialized training in reasoning. For all LLMs, performance generally decreases as instance size increases; however, advanced reasoning models exhibit a constant or substantially less pronounced drop for simple metrics such as feasibility, suggesting that they can maintain awareness of explicit constraints even as problem size

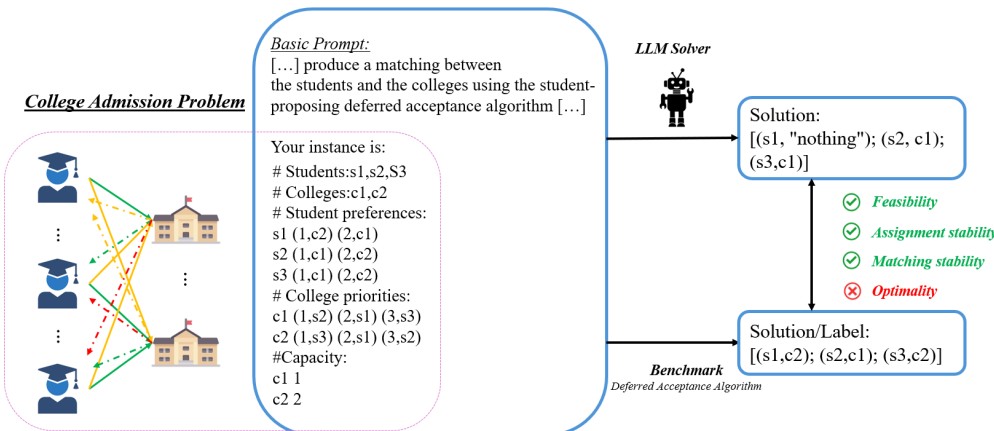

Figure 4: Experimental protocol for generating solutions to the College Admission Problem. We fed the LLM one instance associated with one prompt strategy. We compare the solution to the one of the DA algorithm to measure the feasibility, stability and optimality of the LLM-generated answer.

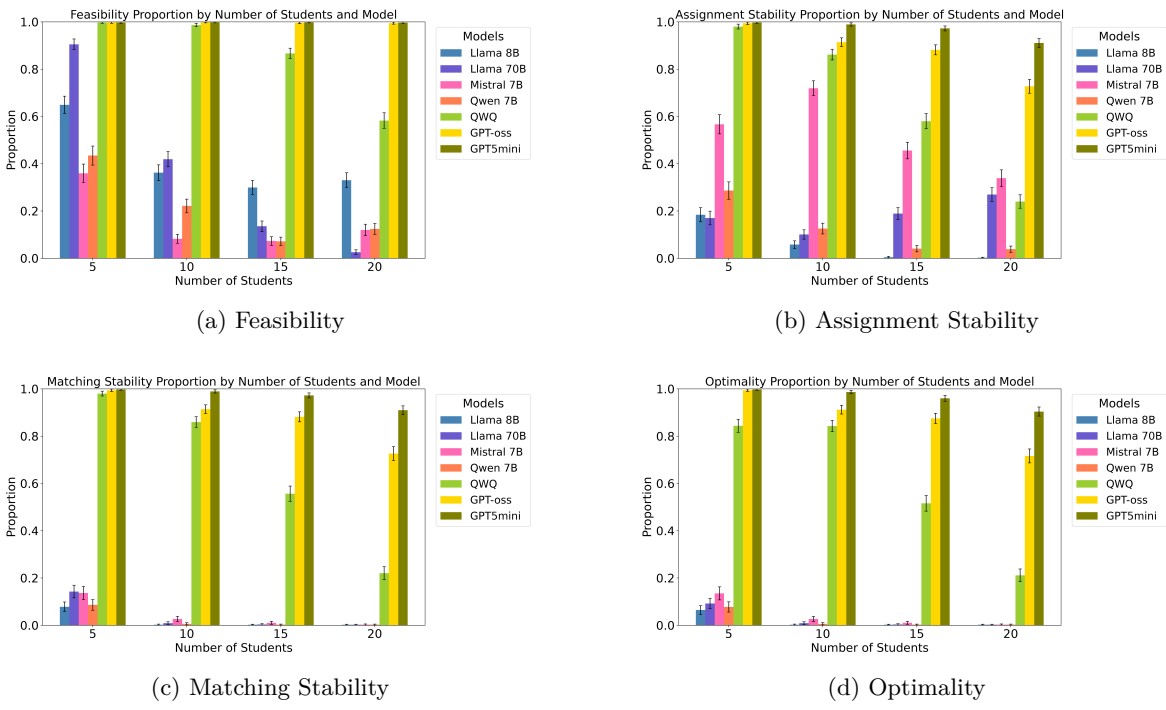

Figure 5: Proportion of correct outcomes by model and number of students.

grows. For more demanding evaluation criteria such as stability and optimality, stronger reasoning models substantially reduce the impact of increasing instance size, although most tested models still show room for improvement on larger instances. Surprisingly, for feasibility, the smaller Llama model yield statistically significantly better results than the larger one, whereas this pattern does not hold for non-explicit metrics such as stability and optimality. A possible hypothesis to explain this behavior is that smaller models prioritize these metrics, whereas larger models may underperform by attempting to satisfy the more demanding ones; however, further testing is needed to confirm this proposed explanation. For stability and optimality, which require understanding and preventing the notion of blocking pairs, the task is more challenging at small scale even for some reasoning models. While the DA algorithm's complexity depends on the number of students,

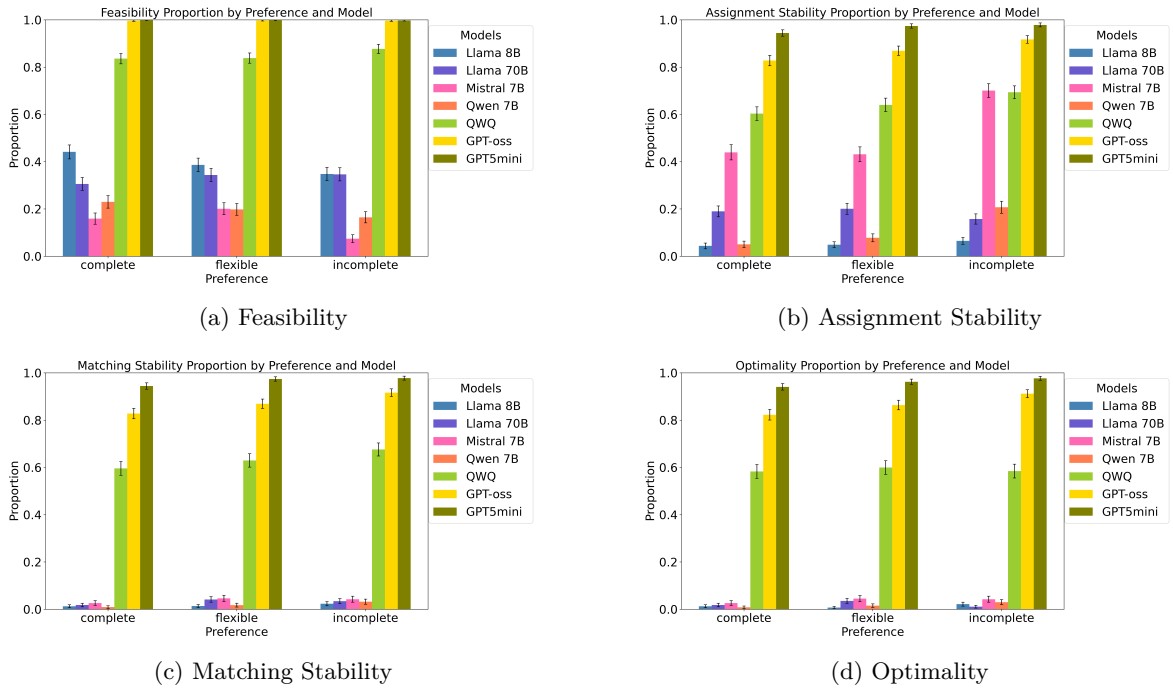

Figure 6: Proportion of correct outcomes by model and preference.

it also depends on the length of the students' preference lists. Figure 6 displays the metric distributions across different preference types and models. Figure 17 in the Appendix E presents the metrics for every prompt strategies instead of models, where the following conclusions can still be drawn. Complete preferences increase the algorithmic complexity by increasing the number of iterations required by the DA to obtain the optimal solution. In contrast, incomplete preferences present more reasoning complexity, as they break completeness, creating more invalid pairs and requiring the algorithm to reason about acceptability and the presence of unmatched participants. In other words, incomplete preferences inherently increase the number of invalid pairs (i.e., the participant would rather remain unmatched than be assigned to a less preferred option). A matching that contains one invalid pair will always be unstable. However, the results suggest that the models do not struggle with this aspect. Performance remains almost unchanged with varying preference types, unlike the number of students, with a slight advantage for incomplete preferences which is only statistically significant for GPT-oss as shown by the confidence interval displayed in Table 7.

**RQ2: How does the prompting strategy and model choice affect the quality of the generated matchings?** We previously mentioned that reasoning LLMs' performance dominates that of the base LLMs, especially when dealing with more complex instances and metrics. Although this shows that specialized training for reasoning is needed to solve even a polynomially solvable problem and understand the core concept underlying matching problems, Figure 5b and Figure 6b reveal that Mistral performed surprisingly well at satisfying the concept of stability, but failed to perform well on the subsequent metrics because it can not satisfy feasibility as well.

In this paper, we tested multiple prompt strategies known to improve reasoning. In Appendix E, Table 4 presents the metrics for each model and prompt. Table 5 and Table 6 present the 95% Wilson CI for prompting performances. For the base models, the gap for a metric between two prompts can reach up to 40%. Although no single prompt achieves statistically significant superiority across all models, some trends emerge from Figure 7. Reasoning models appear to be more consistent in their performance across metrics, and the prompts that perform best for these models differ noticeably from those that perform best under the base models. Interestingly, the General prompt is usually neither the best nor the worst, indicating that omitting the algorithm does not enhance the models' reasoning performance. This possibly reflects that the

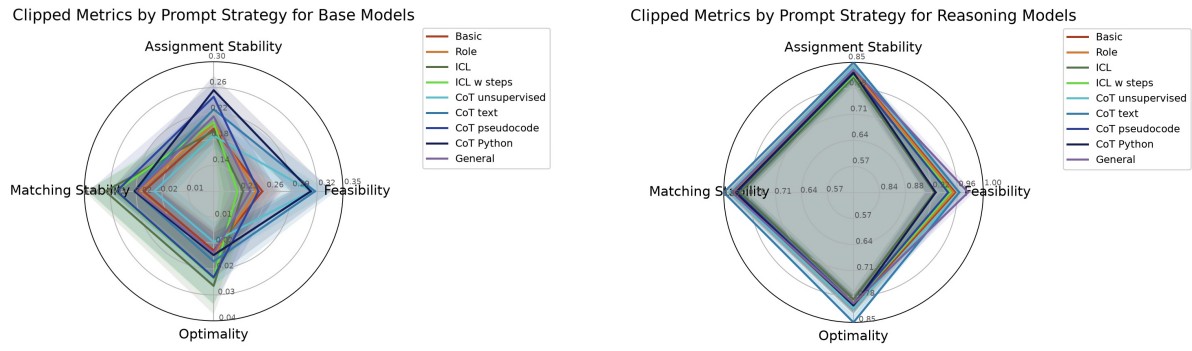

(a) Prompt Performance in Base Models          (b) Prompt Performance in Reasoning Models

Figure 7: Clipped performance by model types and prompts.

models are already aware of the algorithmic procedure needed to produce solutions that meet the required criteria. Ultimately, there is no single golden rule that determines which prompts are associated with higher performance for all models.

**RQ3: Can iterative prompting help LLM achieve higher performances through self-verification and correction?** For the iterative prompting experiments, as we had to respect the token limits of all open-weights LLMs, we restricted our experiments to 5 and 10 students. Table 1 presents the results under role prompting, where the same conclusions can be draw from other prompts under Appendix E with Table 9-16. Additionally, while GPT-oss and GPT 5 mini showed great results previously, the models are more restrictive in the resources, so we limited the testing on the reasoning model to QwQ. In the cases where the model failed to produce the correct answer within the 5 attempts, we select the best attempt, i.e., the one that satisfied the highest number of metrics. Although this could give the impression that LLMs can understand the feedback given on this problem and correct the past errors to some degree, we can show that improvement is not always monotonic by returning the last matching. In this case, while it is often possible to have very similar performance when returning the last or best attempt, there are other cases where it can also decrease significantly. This reveals that iterative prompting is not always iteratively improving the model's behavior and reasoning. Evaluating the impact of self-correctness is challenging for highly capable models, as this can only be measured when a model actually fails within the 5 attempts to compute the student-optimal stable matching. For instance, QwQ sometimes fails on its first try, but it typically reaches the student-optimal stable solution before the iteration limit, making it difficult to observe whether the self-correction process yields monotonic improvements. These results suggest that some models, such as QwQ, can improve their responses through iterative prompting without degradation, whereas the other models evaluated do not consistently exhibit this behavior.

## 5 Discussion, Limitations and Future Work

The results of RQ1 showed that the many-to-one matching problem is a challenging task, where the tested LLMs mostly fail to satisfy the notions of stability, feasibility, and optimality, regardless of the prompts, as the number of students increases. They indicate that certain aspects of the DA algorithm's runtime complexity, such as an increase in the number of students, can be associated with a sharp decline in performance. However, the impact is not the same as the preferences, which also impact the algorithm's complexity and solution space. LLMs do not have a problem with incomplete preferences, which create invalid pairs, but it also does not lead to significantly better performance for most models. One possible hypothesis is that when students are matched early in the iterative process, the distinction between preference types has limited impact, whereas the number of students consistently influences model performance. However, complete preferences may introduce greater difficulty in instances where the student-optimal matching assigns students to colleges ranked among their least preferred options. More experiments are needed to analyze how preference

Table 1: Iterative prompting with Role prompting for instance with 5-10 students. For each model, we have the metrics for no iterative prompting and iterative prompting with the last and best attempt.

| Model | Attempt | Feasibility | Assignment | Matching | Optimality |
|---|---|---|---|---|---|
| Llama 8B | No | 0.4678 (-) | 0.1170 (-) | 0.0409 (-) | 0.0351 (-) |
| Llama 8B | Last | 0.4386 (-2.9%) | 0.1287 (+1.2%) | 0.0760 (+3.5%) | 0.0643 (+2.9%) |
| Llama 8B | Best | 0.6082 (+14.0%) | 0.1579 (+4.1%) | 0.0877 (+4.7%) | 0.0643 (+2.9%) |
| Llama 70B | No | 0.6374 (-) | 0.0877 (-) | 0.0468 (-) | 0.0234 (-) |
| Llama 70B | Last | 0.8655 (+22.81%) | 0.1170 (+2.9%) | 0.1111 (+6.4%) | 0.0819 (+5.6%) |
| Llama 70B | Best | 0.8830 (+24.6%) | 0.1462 (+5.6%) | 0.1404 (+9.4%) | 0.0819 (+5.6%) |
| Mistral 7B | No | 0.2164 (-) | 0.5556 (-) | 0.0585 (-) | 0.0585 (-) |
| Mistral 7B | Last | 0.1930 (-2.3%) | 0.5673 (+1.2%) | 0.0819(+2.3%) | 0.0819(+2.3%) |
| Mistral 7B | Best | 0.2281 (+1.2%) | 0.6140 (+5.8%) | 0.0819 (+2.3%) | 0.0819 (+2.3%) |
| Qwen 7B | No | 0.2398 (-) | 0.2573 (-) | 0.0351 (-) | 0.0351 (-) |
| Qwen 7B | Last | 0.4503 (+21.1%) | 0.0760 (-18.1%) | 0.0526 (+1.8%) | 0.0468 (+1.2%) |
| Qwen 7B | Best | 0.5556 (+31.6%) | 0.1930 (-6.4%) | 0.0585 (+2.3%) | 0.0468 (+1.2%) |
| QwQ 32B | No | 0.9942 (-) | 0.9064 (-) | 0.9006 (-) | 0.8304 (-) |
| QwQ 32B | Last | 1.0000 (+0.6%) | 0.9883 (+8.2%) | 0.9883 (+8.8%) | 0.9474 (+11.7%) |
| QwQ 32B | Best | 1.0000 (+0.6%) | 0.9883 (+8.2%) | 0.9883 (+8.8%) | 0.9474 (+11.7%) |

constraints impact the solution when scaling, or whether they would still lead to the same conclusions if the models were not instructed to follow this algorithm, where invalid pairs are naturally avoided.

Moreover, the results of RQ2 highlight the sensitivity of the models. There is no prompt which systematically offers better performances over all models. More precisely, the performances under a prompting strategy depend on several aspects, including the model used, the corresponding number of parameters it has, and the observed metric. Additionally, even if we attempt to enhance the performance of base LLMs with these strategies, they will still fall short of matching the performance of reasoning LLMs with minimal prompting. However, there is a trend between strategies that are better fitted for base or reasoning LLMs. We hypothesize that base models can benefit from having detailed prompts, which will guide their thought process, making a substantial difference in the final results, while reasoning models have sufficient capabilities to do so with general guidelines. On the other hand, if LLMs are given access to tools, their performance could be more reliable, and prompts including the code would probably work best among all models, including reasoning ones.

Additionally, RQ3 demonstrates that iterative prompting is a promising approach for improving LLMs' reasoning under challenging tasks. However, the fact that results do not always converge monotonically toward a better solution raises questions regarding how effectively LLMs integrate and exploit feedback. Appendix E.5 provide supplementary bar plots presenting the proportion of metrics satisfied across iterations for all models and prompts. These show that there is no clear pattern of systematic degradation or over-correction over successive iterations. Further experiments are therefore needed to better understand how feedback influences model reasoning and performance, for example by analyzing which iterations most frequently yield the best results, or by studying how different forms and levels of feedback detail affect the refinement process.

While this work provides a valuable benchmark for a previously underexplored setting, it still has some limitations. Although testing each strategy allowed us to observe its individual impact on performance, further improvements may be achieved by combining strategies or by extending our designs to include multiple examples in ICL. A common challenge in evaluating open-weight LLMs is fitting the prompt and output within the limit of allowed tokens, which also prevent us from scaling the instances to larger sizes. For closed-source models, the model evaluated in this study shows promising results, suggesting an improved understanding of key concepts of the CAP. However, assessing whether such performance can be maintained on larger instances would require substantially more computational resources, increasing evaluation costs and potentially limiting their accessibility for broader and extensive experimentation. While realistic instances

of this class of problems typically involve a larger number of students, and it would have been of interest to evaluate the limits of each model beyond the instance sizes considered in this paper, our experiments nevertheless demonstrate that even with up to 20 students, the models exhibit limitations in their reasoning capabilities when solving matching problems. Base models are unable to satisfy complex metrics for any instance sizes, showing that there is absolutely no understanding of those notions, but reasoning models do understand those at very small scales. Even though LLMs might not be used to directly solve this problem given the existence of the DA algorithm, these findings raise concerns about the use of such models for heuristic generation in NP-hard combinatorial optimization problems, as they already exhibit significant difficulties when addressing substantially simpler tasks. Another limitation concerns the models tested in this article. Although other LLMs have achieved stronger performance on reasoning tasks, we limited our study to models that can be used with reasonable resources. While future work could expand the range of models tested, this work serves as an initial step toward identifying LLM models strengths and weaknesses when responding to optimization problems with complex constraints, such as stability and preference based ones. The DA algorithm is the standard and most widely used mechanism in many-to-one matching markets. It guarantees stability and strategy-proof (i.e., students have incentive to report their true preferences). However, DA does not guarantee Pareto optimality and, despite running in polynomial time, is considered inefficient (Ortega & Klein, 2023). Alternative mechanisms, such as Top Trading Cycles (TTC) and Risk Minimization (RM), prioritize different objectives. In particular, RM does not ensure stability or strategy-proofness, but favors greater efficiency and more egalitarian rank distributions (Ortega & Klein, 2023). Our results indicate that current LLMs struggle to reliably implement procedures such as DA, and omitting the algorithm does not improve performance. A promising direction for future research is therefore not to position LLMs as direct solvers of these mechanisms, but rather as tools that could assist practitioners in selecting the mechanism that best corresponds to the specific priorities of the market at hand.

## 6 Conclusions

Solving a two-sided many-to-one matching problem remains challenging for open-weight LLMs. Although an algorithm exists that can compute a feasible, stable, and student-optimal solution in polynomial time, ensuring that the outputs of the models respect these properties is far from straightforward. Our benchmark reveals how well those models handle structured preference-based reasoning, providing direct insight into their ability to align with domain-specific constraints. Specific properties, such as achieving a student-optimal stable matching, are particularly difficult to attain when models are not recent frontier closed-source models. Our iterative prompting experiments highlight the core challenge of self-correction based on feedback. We also observed that the models are more significantly affected by the number of students than by the type of preferences, even though both factors impact the running time of the DA algorithm. This suggests that the models struggle to follow the iterative structure required by this polynomial-time procedure correctly. Reasoning LLMs consistently outperform base ones across all metrics. Surprisingly, Mistral shows strong performance on stability despite being a base model. While no prompting strategy is universally optimal across all models, each model has a CoT variant that performs reliably well, making it a broadly effective approach.

## Acknowledgments

This work was funded by the Canada CIFAR AI Chair, the CIFAR Catalyst award, the FRQNT Master Scholarship No. B1X-341759, IVADO, the NSERC Canada Graduate Research Scholarship, and the NSERC Grant No. 2024-04051. We also thank Compute Canada and Mila clusters for their support in providing facilities for our evaluations.

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

## A Related Work

### A.1 Reasoning over Preference in agent systems

Reasoning over users' preferences has been widely studied in game theory and economics, where the behaviors of all agents are encoded into their preferences (Stirling & Felin, 2013). Artificial Intelligence (AI) has also been interested in this concept for many years (Rossi et al., 2011; Walsh, 2007). In the College Admission Problem, we have two sets of participants, and each participant in one set expresses its preferences over the participants in the other set. Alternatively, recent works in multi-agent systems represent the preference of each agent over a set of objects in a qualitative way (Rossi et al., 2011). One prominent approach in this line of research is the use of Conditional Preference networks (CP-nets) to have a compact, qualitative, and graphical representation of preferences (Boutilier et al., 2004). Several extensions of CP-nets have been proposed to better capture complex reasoning settings. (Rossi et al., 2004) introduced mCP nets, a set of partial CP-nets, to represent the preferences of multiple agents, while PCP nets introduce uncertainty with their probabilistic framework (Cornelio et al., 2021). In a related line of research, scholars have studied reasoning over preferences in Belief-Desire-Intention (BDI) agent systems (Visser et al., 2011). In contrast to compact and often partial preference formalisms like CP nets, the classic College Admission Problem assumes that participants report strict orderings, either complete or not, which simplifies representation but still requires non-trivial reasoning over constraints and preferences. Recently, LLM-based agents within a multi-agent system have been used to capture user preferences and even align themselves in recommender systems (Zhang et al., 2023a). In this paper, we use an LLM as a centralized reasoning solver, where it must interpret the explicitly provided preferences of individual participants in the College Admission Problem and simulate the underlying algorithmic reasoning required to produce feasible, stable, and student-optimal matchings.

## B Instances and Prompts

An instance of the College Admission Problem consists of students and colleges that must be written in a specific format to be compatible with the DA algorithm. Therefore, we introduce in the following Figure 1

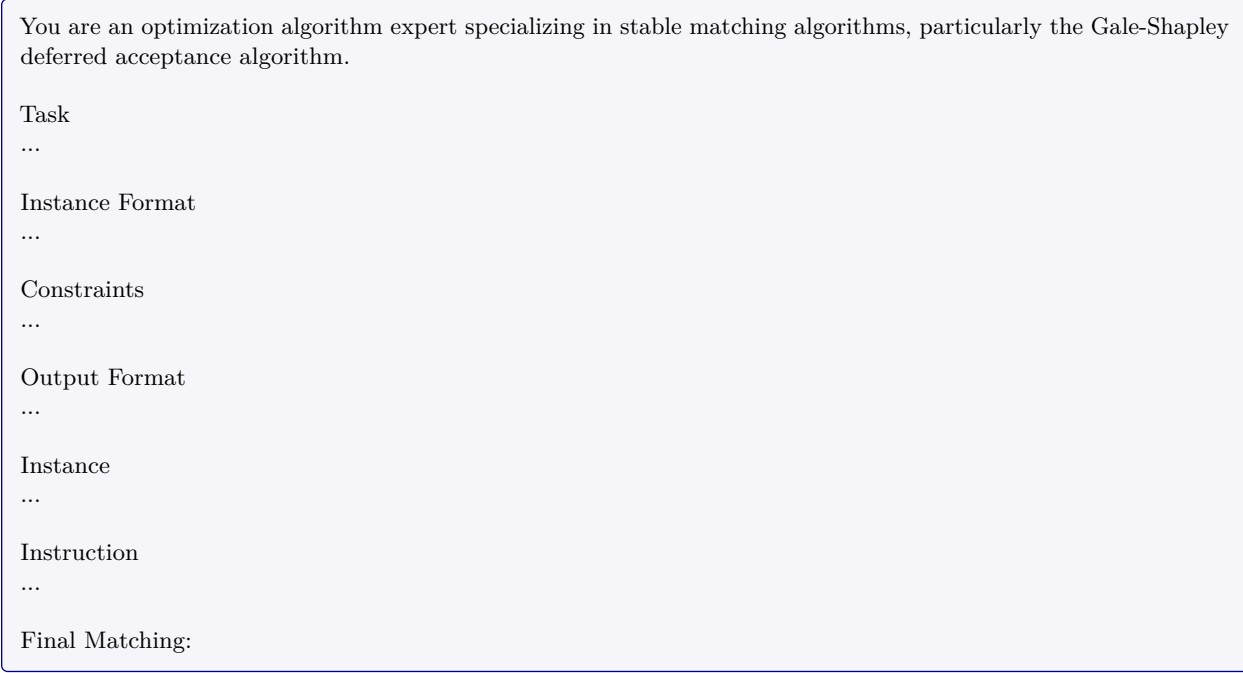

You are an optimization algorithm expert specializing in stable matching algorithms, particularly the Gale-Shapley deferred acceptance algorithm.

Task
...

Instance Format
...

Constraints
...

Output Format
...

Instance
...

Instruction
...

Final Matching:

Figure 8: Template of our **Role** prompt strategy

an example of such an instance. Specifically, each instance consists of the number of students and colleges, followed by a list stating their names. The capacities of each school will follow it, and then we will have the preferences of students and colleges, respectively. As we compare the performance of the output of LLMs with the results of the DA algorithm, we compile the matching given by the DA algorithm for this prompt. The Figure 2 provides the answer format, as we expect it to be returned by LLM models. Note that when a student is matched to "nothing", it simply means that this student is unassigned. There is a pair for each student in the matching, but it is not necessarily for each school if no students are assigned to that school.

As detailed in Section 3.2, our benchmark evaluates model performance across four levels of student population, three types of preference, 3 different capacity level, four ratios and three random seeds. Certain combinations are excluded due to infeasibility, like under-capacity settings with a 1:1 ratio, which would imply fewer seats than schools, which is not meaningful in this context. The final dataset comprises 369 total instances per model–prompt combination. Specifically, for each preference type, there are 123 instances. For each model-prompt combination, we have 99 instances corresponding to 10,15 or 20 students, while there are 72 for 5 students. This design enables fine-grained analysis across different parameter settings while ensuring a sufficient number of instances to support statistically meaningful conclusions.

Each of the 369 benchmark instances was evaluated across six language models and eight distinct prompting strategies, as described in Section 3.3. Following conventions established in prior work (Juneja et al., 2025; Liu et al., 2025), each prompt was structured into distinct components, with explicit headers indicating the purpose of each section. We also appended the phrase "Final Matching:" at the end of each prompt to explicitly signal the model to return the final assignment. This technique, which is commonly used in question-answering tasks, helps improve output validity (Zhang et al., 2023b). The prompts are presented from Figure 8 to Figure 14, with redundant content omitted for brevity.

**Iterative Prompting Feedback:** For iterative prompting, we are providing general feedback on the solution returned by the LLM. More specifically, if there is a matching in the previous output, we will return the feedback following this example that was feasible and stable, but not optimal. Figure 15 shows an example of how such feedback is written.

Task
...

Instance Format
...

Constraints
...

Output Format
...

Example
Below is a solved example demonstrating input format and expected output. [ADD EXAMPLE HERE]

Instance
...

Instruction
...

Final Matching:

Figure 9: Template of our **ICL** prompt strategy

Table 2: Proportion of output, out of the 369 possibilities, that generate an acceptable matching (i.e a matching that we could extract from the output)

|                  | Llama 8B | Llama 70B | Mistral | Qwen   | QwQ    | GPT-oss | GPT 5 mini |
|------------------|----------|-----------|---------|--------|--------|---------|------------|
| Basic            | 1.0000   | 1.0000    | 0.9837  | 1.0000 | 0.9892 | 0.9973  | 1.0000     |
| Vague            | 1.0000   | 1.0000    | 0.9919  | 0.9783 | 0.9973 | 1.0000  | 1.0000     |
| Role             | 1.0000   | 1.0000    | 0.9864  | 1.0000 | 0.9973 | 1.0000  | 1.0000     |
| ICL              | 1.0000   | 1.0000    | 0.7859  | 1.0000 | 0.9783 | 1.0000  | 1.0000     |
| ICL w steps      | 0.9458   | 1.0000    | 0.9729  | 1.0000 | 0.9729 | 1.0000  | 1.0000     |
| CoT unsupervised | 1.0000   | 1.0000    | 0.9566  | 0.9973 | 1.0000 | 1.0000  | 1.0000     |
| CoT text         | 1.0000   | 1.0000    | 0.9837  | 0.8808 | 0.9946 | 1.0000  | 1.0000     |
| CoT pseudocode   | 1.0000   | 1.0000    | 0.9702  | 0.9864 | 0.9946 | 0.9946  | 1.0000     |
| CoT Python       | 1.0000   | 1.0000    | 0.9675  | 1.0000 | 0.9837 | 1.0000  | 0.9946     |

## C   Valid Output

We defined validity in Section 4. Table 2 reports the proportion of the 369 instances that are valid. Invalid outputs include cases where the generated matching contains incorrect names (e.g., referring to students as "Alice" and "Bob" instead of s1 and s2), or when the model returns code rather than a matching. As shown in Table 2, there is no major problem and the proportions of valid output are almost always close to 1. However, Mistral exhibits a comparatively higher rate of invalid outputs, particularly when using the ICL with steps prompt. Qwen also produces a smaller number of invalid outputs under certain CoT prompting strategies. In contrast, both Llama models consistently generate valid outputs across all instances. It is worth noting that most invalid responses occurred for larger instances, suggesting that model can have a harder time following the instruction with bigger instances.

Task

...

Instance Format

...

Constraints

...

Output Format

...

Example

Here is an example consisting of an instance, the step-by-step application of the algorithm, and the final matching.

**Instance**

# Num. students: 5

# Num. Colleges: 3

# Students: s1, s2, s3, s4, s5

# Colleges: c1, c2, c3

# Capacities:

c1 2,

c2 1,

c3 2

# Student preferences:

s1 (1,c1) (2,c2) (3,c3)

s2 (1,c1) (2,c3) (3,c2)

s3 (1,c2) (2,c1) (3,c3)

s4 (1,c3) (2,c2) (3,c1)

s5 (1,c1) (2,c2) (3,c3)

# Colleges priorities:

c1 (1,s2) (2,s1) (3,s5) (4,s3) (5,s4)

c2 (1,s3) (2,s1) (3,s4) (4,s5) (5,s2)

c3 (1,s4) (2,s2) (3,s5) (4,s1) (5,s3)

**Step by Step DA Algorithm**

Round 1 :

       s1, s2, s3, s4, s5 propose to their 1st choices, c1, c1, c2, c3, c1 respectively.

       c1 (capacity of 2) gets proposals from s1, s2, s5. It prefers s1 and s2. It rejects s5.

       c2 (capacity of 1) gets a proposal from s3 and holds it

       c3 (capacity of 2) gets a proposal from s4 and holds it.

       Applications of hold : c1 :s1,s2, c2 :s3 and c3:s4. Rejected:s5

Round 2 :

       s5 (previously rejected) proposes to its next choice, c2.

       c2 (capacity of 1) compares the new proposal (s5) to its current hold (s3). It prefers s3. It rejects s5.

       Applications of hold : c1 :s1,s2, c2 :s3 and c3:s4. Rejected:s5

Round 3 :

       s5 (previously rejected) proposes to its next choice, c2.

       c3 (capacity of 2) gets a proposal from s5. It is holding s4 and has an open spot. It holds s5.

       Applications of hold : c1 :s1,s2, c2 :s3 and c3:s4,s5. Rejected:

There was no rejection on the last round, the algorithm ends and the colleges accept the students on hold.

**Final Matching**

The final matching is :

(s1,c1),(s2,c1),(s3,c2),(s4,c3),(s5,c3)

Instance

...

Instruction

...

Final Matching:

Figure 10: Template of our **ICL w steps** prompt strategy

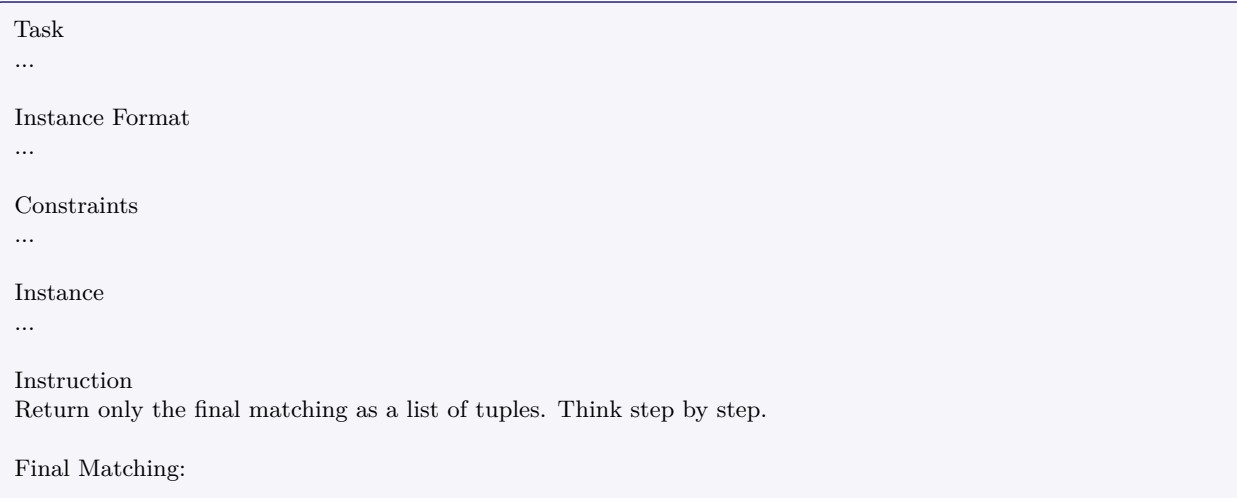

Task
...

Instance Format
...

Constraints
...

Instance
...

Instruction
Return only the final matching as a list of tuples. Think step by step.

Final Matching:

Figure 11: Template of our **CoT unsupervised** prompt strategy

Task
...

Instance Format
...

Constraints
...

Output Format
...

Instance
...

Step by Step Process

1. Each student applies to their favorite college.
2. Each college rejects all applications from students that are unacceptable to it. If a college received at most q applications from acceptable students so far, all those students are put on the colleges waiting list. Otherwise the college puts its favorite q students among all applicants on the waiting list and rejects all remaining ones.
3. Each student that was rejected in the previous step applies to his favorite among the colleges he or she has not yet applied to.
4. Steps 2 and 3 are repeated until it holds for all students that they were either not rejected in the previous step or already applied to all colleges acceptable to them.
5. Each college admits all students on its waiting list.

Instruction
...

Final Matching:

Figure 12: Template of our **text CoT** prompt strategy

Task
...

Instance Format
...

Constraints
...

Output Format
...

Instance
...
Step by Step Process

```
# INITIALIZE:
for each student s in S:
  s.next_choice ← 1          # index into Pref(s), starting at top choice
  s.is_free     ← true
for each college c in C:
  c.waiting_list ← ∅         # no one tentatively held yet
  c.applicants   ← ∅         # applications in the current round

REPEAT
  # Free students apply to their next choice
  for each student s in S:
    if s.is_free and s.next_choice ≤ length(Pref(s)):
      let c = Pref(s)[s.next_choice]
      add s to c.applicants
      s.next_choice ← s.next_choice + 1

  # Colleges review their applicant pools
  any_rejection ← false
  for each college c in C:
    pool ← c.waiting_list ∪ c.applicants
    remove from pool any student unacceptable to c
    sort pool by c's priority order (best first)
    c.waiting_list ← first q(c) students in pool
    rejected ← pool minus c.waiting_list
    for each student r in rejected:
      r.is_free ← true
      any_rejection ← true
    for each student a in c.waiting_list:
      a.is_free ← false
    c.applicants ← ∅

UNTIL any_rejection is false

# Final assignments
for each college c in C:
  admit every student in c.waiting_list
```

Instruction
...

Final Matching:

Figure 13: Template of our **pseudocode CoT** prompt strategy

Task
...

Instance Format
...

Constraints
...

Output Format
...

Instance
...

Step by Step Process

```python
priority_ranks = {
    c: {s: rank for rank, s in enumerate(priority)}
    for c, priority in college_priorities.items()
}

proposals = {s: deque(prefs) for s, prefs in student_prefs.items()}
college_matches = defaultdict(list)
student_match = {s: None for s in students}

while True:
    free_students = [
        s for s in students
        if student_match[s] is None and proposals[s]
    ]
    if not free_students:
        break

    for student in free_students:
        college = proposals[student].popleft()
        college_matches[college].append(student)

        accepted = sorted(
            college_matches[college],
            key=lambda s: priority_ranks[college][s]
        )[:capacities[college]]

        for s in college_matches[college]:
            if s in accepted:
                student_match[s] = college
            else:
                student_match[s] = None

        college_matches[college] = accepted
```

Instruction
...

Final Matching:

Figure 14: Template of our **Python CoT** prompt strategy

> Attempted Response:
>
> Previous matching include here
>
> This matching is incorrect. Here is some feedback based on four metrics:
>
> - Feasibility: The matching is feasible. (number of seats over capacity= 0)
>
> - Assignment stability: The matching is assignment stable. (number of blocking pairs=0)
>
> - Matching stability: The matching is matching stable. (number of blocking pairs=0)
>
> - Student-optimality: The matching is not student optimal. Some students do not have their best matching of all stable outcomes. (proportion of correct pairs= 0.8)

Figure 15: Feedback template example for a feasible and stable, but not student-optimal matching with iterative prompting.

## D  Inference Configuration

For reproducibility purposes, the configuration used during inference generation is presented here. All models were used with Transformers version 4.47.6, but vLLM version 0.17.1 was later tested to see if it would significantly improve the results, which was not the case. The hardware used was 4 H100 80GB GPUs for GPT-oss 120B and 2 H100 80GB GPUs for all other models. MXFP4 quantization was only used for GPT-oss 120B, as it significantly improved performance over the fallback option of bfloat16. The list of commit for each model is:

- Llama 3 8B: 8afb486c1db24fe5011ec46dfbe5b5dccdb575c2

- Llama 3 70B: 6f6073b423013f6a7d4d9f39144961bfbfbc386b

- Mistral 7B: ec5deb64f2c6e6fa90c1abf74a91d5c93a9669ca

- Qwen2 7B: f2826a00ceef68f0f2b946d945ecc0477ce4450c

- QwQ 32B: 976055f8c83f394f35dbd3ab09a285a984907bd0

- GPT oss 120B: 8b193b0ef83bd41b40eb71fee8f1432315e02a3e.

- GPT 5 mini: gpt-5-mini-2025-08-07

The environment is provided with our code under footnote 1.

## E  Tables and Graphs

### E.1  Time Table

We present in Table 3 the time each LLM took in order to generate the output of 10 random instances across our datasets, making sure we had similar numbers for different complexity. Aggregating over the number of students instead, we can see that the number of students is also impacting the inference time, with an average of 53.4, 102.6, 140.1 and 183.8 seconds for a respective number of 5,10,15 and 20 students.

Table 3: Average time (in seconds) to generate the output for the College Admission Problem across 10 instances.

| Model | Llama 8B | Llama 70B | Mistral 7B | Qwen 7B | QwQ 32B | GPT-oss 120B |
|---|---|---|---|---|---|---|
| Average Time | 14.3111 | 6.9977 | 1.7036 | 6.5190 | 391.7150 | 324.5929 |
| Standard deviation | 5.347 | 2.918 | 0.9432 | 4.930 | 129.0224 | 308.9703 |

### E.2 Aggregated Tables

We include in this section, under Table 4 the aggregated results across the 369 instances, when are of them are valid, for each prompt strategy and model combination. While our figures included in the paper represent very well the data by number of students and preference types, this table allows to have a more broad view of the overall performances.

Table 4: Overall performances over our 4 metrics for all model and prompt. For brevity, Assignment means Assignment Stability, Matching means Matching Stability, CoT unsup. refers to CoT unsupervised and CoT pseudo refers to CoT pseudocode.

| Prompt | Model | Count | Feasibility | Assignment | Matching | Optimality |
|---|---|---|---|---|---|---|
| Basic | Llama8B | 369 | 0.3496 | 0.0542 | 0.0163 | 0.0136 |
| Basic | Llama70B | 369 | 0.3550 | 0.1463 | 0.0244 | 0.0136 |
| Basic | Mistral | 363 | 0.1460 | 0.4793 | 0.0303 | 0.0303 |
| Basic | Qwen | 369 | 0.1707 | 0.1084 | 0.0108 | 0.0108 |
| Basic | QwQ | 365 | 0.8750 | 0.6467 | 0.6332 | 0.5788 |
| Basic | GPT-oss | 368 | 1.0000 | 0.8832 | 0.8832 | 0.8804 |
| Basic | GPT 5 mini | 369 | 1.0000 | 0.9594 | 0.9594 | 0.9594 |
| General | Llama8B | 369 | 0.2873 | 0.0786 | 0.0244 | 0.0136 |
| General | Llama70B | 369 | 0.3577 | 0.1491 | 0.0217 | 0.0136 |
| General | Mistral | 366 | 0.1311 | 0.5109 | 0.0301 | 0.0301 |
| General | Qwen | 361 | 0.1884 | 0.1219 | 0.0139 | 0.0139 |
| General | QwQ | 368 | 0.9375 | 0.6984 | 0.6902 | 0.6304 |
| General | GPT-oss | 369 | 0.9973 | 0.8455 | 0.8455 | 0.8374 |
| General | GPT 5 mini | 369 | 1.0000 | 0.9512 | 0.9512 | 0.9322 |
| Role | Llama8B | 369 | 0.3523 | 0.0542 | 0.0190 | 0.0163 |
| Role | Llama70B | 369 | 0.3360 | 0.1680 | 0.0217 | 0.0108 |
| Role | Mistral | 364 | 0.1593 | 0.4560 | 0.0302 | 0.0302 |
| Role | Qwen | 369 | 0.1463 | 0.1409 | 0.0163 | 0.0163 |
| Role | QwQ | 368 | 0.8750 | 0.6467 | 0.6332 | 0.5788 |
| Role | GPT-oss | 369 | 0.9946 | 0.8645 | 0.8645 | 0.8537 |
| Role | GPT 5 mini | 369 | 0.9973 | 0.9702 | 0.9675 | 0.9645 |
| ICL | Llama8B | 369 | 0.2087 | 0.0867 | 0.0271 | 0.0217 |
| ICL | Llama70B | 369 | 0.3306 | 0.2087 | 0.0407 | 0.0298 |
| ICL | Mistral | 290 | 0.1897 | 0.4586 | 0.0448 | 0.0448 |
| ICL | Qwen | 369 | 0.1951 | 0.0705 | 0.0190 | 0.0190 |
| ICL | QwQ | 361 | 0.7756 | 0.5900 | 0.5789 | 0.5319 |
| ICL | GPT-oss | 369 | 1.0000 | 0.8591 | 0.8591 | 0.8537 |
| ICL | GPT 5 mini | 369 | 1.0000 | 0.9783 | 0.9783 | 0.9756 |
| ICL w steps | Llama8B | 349 | 0.3352 | 0.0201 | 0.0057 | 0.0057 |
| ICL w steps | Llama70B | 369 | 0.2791 | 0.2493 | 0.0407 | 0.0298 |

Continued on next page

| Prompt | Model | Count | Feasibility | Assignment | Matching | Optimality |
|---|---|---|---|---|---|---|
| ICL w steps | Mistral | 359 | 0.1086 | 0.4680 | 0.0446 | 0.0418 |
| ICL w steps | Qwen | 369 | 0.1870 | 0.0813 | 0.0244 | 0.0244 |
| ICL w steps | QwQ | 359 | 0.8524 | 0.5989 | 0.5989 | 0.5599 |
| ICL w steps | GPT-oss | 369 | 1.0000 | 0.8808 | 0.8808 | 0.8808 |
| ICL w steps | GPT 5 mini | 369 | 1.0000 | 0.9648 | 0.9648 | 0.9594 |
| CoT unsup. | Llama8B | 369 | 0.4038 | 0.0461 | 0.0136 | 0.0108 |
| CoT unsup. | Llama70B | 369 | 0.3604 | 0.1653 | 0.0190 | 0.0108 |
| CoT unsup. | Mistral | 353 | 0.1728 | 0.4674 | 0.0255 | 0.0255 |
| CoT unsup. | Qwen | 368 | 0.3071 | 0.0734 | 0.0109 | 0.0109 |
| CoT unsup. | QwQ | 369 | 0.8943 | 0.6748 | 0.6612 | 0.6070 |
| CoT unsup. | GPT-oss | 368 | 1.0000 | 0.8750 | 0.8750 | 0.8723 |
| CoT unsup. | GPT 5 mini | 369 | 1.0000 | 0.9675 | 0.9675 | 0.9621 |
| CoT text | Llama8B | 369 | 0.5989 | 0.0244 | 0.0054 | 0.0054 |
| CoT text | Llama70B | 369 | 0.3198 | 0.1843 | 0.0325 | 0.0217 |
| CoT text | Mistral | 363 | 0.1515 | 0.5289 | 0.0331 | 0.0331 |
| CoT text | Qwen | 325 | 0.1785 | 0.1631 | 0.0215 | 0.0215 |
| CoT text | QwQ | 367 | 0.8965 | 0.7411 | 0.7248 | 0.6866 |
| CoT text | GPT-oss | 369 | 0.9973 | 0.9051 | 0.9024 | 0.8970 |
| CoT text | GPT 5 mini | 369 | 1.0000 | 0.9837 | 0.9837 | 0.9810 |
| CoT pseudo | Llama8B | 369 | 0.3821 | 0.0379 | 0.0081 | 0.0081 |
| CoT pseudo | Llama70B | 369 | 0.3252 | 0.1463 | 0.0379 | 0.0271 |
| CoT pseudo | Mistral | 358 | 0.1257 | 0.6508 | 0.0447 | 0.0447 |
| CoT pseudo | Qwen | 364 | 0.1676 | 0.1566 | 0.0247 | 0.0220 |
| CoT pseudo | QwQ | 367 | 0.8420 | 0.6512 | 0.6349 | 0.5995 |
| CoT pseudo | GPT-oss | 367 | 1.0000 | 0.8719 | 0.8719 | 0.8638 |
| CoT pseudo | GPT 5 mini | 369 | 1.0000 | 0.9404 | 0.9404 | 0.9377 |
| CoT python | Llama8B | 369 | 0.6016 | 0.0542 | 0.0136 | 0.0136 |
| CoT python | Llama70B | 369 | 0.3198 | 0.2168 | 0.0271 | 0.0190 |
| CoT python | Mistral | 357 | 0.1036 | 0.6723 | 0.0392 | 0.0392 |
| CoT python | Qwen | 369 | 0.2168 | 0.0921 | 0.0136 | 0.0027 |
| CoT python | QwQ | 363 | 0.7851 | 0.6088 | 0.5950 | 0.5592 |
| CoT python | GPT-oss | 369 | 0.9973 | 0.8699 | 0.8699 | 0.8672 |
| CoT python | GPT 5 mini | 368 | 1.0000 | 0.9891 | 0.9891 | 0.9864 |

## E.3 95% Confidence Interval Tables

While Figure 7a-7b present the performance of each model under different prompts, it is not representing the confidence of our results. Table 5 presents the results, which are not normalized, along with the 95% Wilson confidence interval aggregated over all models. Table 6 presents the same results, but for all prompts and models. Table 5 shows that for each metric, although a prompt may have the highest lower bound, there is always at least one other prompt whose upper bound exceeds it, indicating that differences are not statistically significant.

Table 6: 95% Wilson confidence interval for performance by prompts and models

| Prompt | Model | Feasibility | Assignment | Matching | Optimality |
|---|---|---|---|---|---|
| Basic | Llama 8B | [0.3027, 0.3996] | [0.0354, 0.0822] | [0.0075, 0.0350] | [0.0058, 0.0313] |
| | Llama 70B | [0.3079, 0.4051] | [0.1139, 0.1860] | [0.0129, 0.0457] | [0.0058, 0.0313] |
| | Mistral 7B | [0.1134, 0.1860] | [0.4284, 0.5307] | [0.0170, 0.0534] | [0.0170, 0.0534] |

| Prompt | Model | Feasibility | Assignment | Matching | Optimality |
|---|---|---|---|---|---|
| | Qwen 7B | [0.1358, 0.2125] | [0.0806, 0.1442] | [0.0042, 0.0275] | [0.0042, 0.0275] |
| | QWQ | [0.8360, 0.9042] | [0.5962, 0.6939] | [0.5823, 0.6807] | [0.5269, 0.6277] |
| | GPT-oss | [0.9897, 1.0000] | [0.8463, 0.9121] | [0.8463, 0.9121] | [0.8433, 0.9097] |
| | GPT 5 mini | [0.9897, 1.0000] | [0.9340, 0.9752] | [0.9340, 0.9752] | [0.9275, 0.9710] |
| General | Llama 8B | [0.2435, 0.3354] | [0.0553, 0.1106] | [0.0129, 0.0457] | [0.0058, 0.0313] |
| | Llama 70B | [0.3105, 0.4079] | [0.1163, 0.1890] | [0.0110, 0.0422] | [0.0058, 0.0313] |
| | Mistral | [0.1004, 0.1696] | [0.4599, 0.5618] | [0.0169, 0.0530] | [0.0169, 0.0530] |
| | Qwen | [0.1514, 0.2319] | [0.0921, 0.1597] | [0.0059, 0.0320] | [0.0059, 0.0320] |
| | QWQ | [0.9080, 0.9580] | [0.6496, 0.7430] | [0.6412, 0.7353] | [0.5800, 0.6782] |
| | GPT-oss | [0.9848, 0.9995] | [0.8051, 0.8788] | [0.8051, 0.8788] | [0.7963, 0.8715] |
| | GPT 5 mini | [0.9897, 1.0000] | [0.9242, 0.9689] | [0.9242, 0.9689] | [0.9019, 0.9537] |
| Role | Llama 8B | [0.3053, 0.4023] | [0.0354, 0.0822] | [0.0092, 0.0386] | [0.0075, 0.0350] |
| | Llama 70B | [0.2898, 0.3857] | [0.1333, 0.2095] | [0.0110, 0.0422] | [0.0042, 0.0275] |
| | Mistral 7B | [0.1253, 0.2005] | [0.4056, 0.5074] | [0.0170, 0.0533] | [0.0170, 0.0533] |
| | Qwen 7B | [0.1139, 0.1860] | [0.1091, 0.1801] | [0.0075, 0.0350] | [0.0075, 0.0350] |
| | QWQ | [0.8373, 0.9050] | [0.5966, 0.6938] | [0.5828, 0.6808] | [0.5278, 0.6282] |
| | GPT-oss | [0.9805, 0.9985] | [0.8258, 0.8957] | [0.8258, 0.8957] | [0.8140, 0.8861] |
| | GPT 5 mini | [0.9848, 0.9995] | [0.9474, 0.9833] | [0.9440, 0.9813] | [0.9407, 0.9793] |
| ICL | Llama 8B | [0.1703, 0.2530] | [0.0621, 0.1199] | [0.0148, 0.0492] | [0.0110, 0.0422] |
| | Llama 70B | [0.2846, 0.3802] | [0.1703, 0.2530] | [0.0248, 0.0660] | [0.0167, 0.0526] |
| | Mistral 7B | [0.1487, 0.2387] | [0.4022, 0.5161] | [0.0264, 0.0752] | [0.0264, 0.0752] |
| | Qwen 7B | [0.1579, 0.2386] | [0.0485, 0.1012] | [0.0092, 0.0386] | [0.0092, 0.0386] |
| | QWQ | [0.7298, 0.8156] | [0.5386, 0.6396] | [0.5274, 0.6288] | [0.4803, 0.5827] |
| | GPT-oss | [0.9897, 1.0000] | [0.8199, 0.8909] | [0.8199, 0.8909] | [0.8140, 0.8861] |
| | GPT 5 mini | [0.9897, 1.0000] | [0.9578, 0.9890] | [0.9578, 0.9890] | [0.9543, 0.9871] |
| ICL w steps | Llama 8B | [0.2877, 0.3863] | [0.0097, 0.0408] | [0.0016, 0.0207] | [0.0016, 0.0207] |
| | Llama 70B | [0.2358, 0.3270] | [0.2079, 0.2959] | [0.0248, 0.0660] | [0.0167, 0.0526] |
| | Mistral 7B | [0.0805, 0.1451] | [0.4170, 0.5196] | [0.0276, 0.0712] | [0.0255, 0.0678] |
| | Qwen 7B | [0.1505, 0.2299] | [0.0575, 0.1137] | [0.0129, 0.0457] | [0.0129, 0.0457] |
| | QWQ | [0.8119, 0.8853] | [0.5474, 0.6483] | [0.5474, 0.6483] | [0.5082, 0.6103] |
| | GPT-oss | [0.9897, 1.0000] | [0.8437, 0.9100] | [0.8437, 0.9100] | [0.8437, 0.9100] |
| | GPT 5 mini | [0.9897, 1.0000] | [0.9407, 0.9793] | [0.9407, 0.9793] | [0.9340, 0.9752] |
| CoT unsupervised | Llama 8B | [0.3550, 0.4546] | [0.0290, 0.0725] | [0.0058, 0.0313] | [0.0042, 0.0275] |
| | Llama 70B | [0.3131, 0.4106] | [0.1309, 0.2066] | [0.0092, 0.0386] | [0.0042, 0.0275] |
| | Mistral 7B | [0.1369, 0.2157] | [0.4160, 0.5195] | [0.0135, 0.0477] | [0.0135, 0.0477] |
| | Qwen 7B | [0.2621, 0.3560] | [0.0509, 0.1046] | [0.0042, 0.0276] | [0.0042, 0.0276] |
| | QWQ | [0.8588, 0.9217] | [0.6254, 0.7206] | [0.6115, 0.7077] | [0.5564, 0.6555] |
| | GPT-oss | [0.9897, 1.0000] | [0.8373, 0.9050] | [0.8373, 0.9050] | [0.8343, 0.9026] |
| | GPT 5 mini | [0.9897, 1.0000] | [0.9440, 0.9813] | [0.9440, 0.9813] | [0.9373, 0.9773] |
| CoT text | Llama 8B | [0.5481, 0.6477] | [0.0129, 0.0457] | [0.0015, 0.0195] | [0.0015, 0.0195] |
| | Llama 70B | [0.2743, 0.3690] | [0.1480, 0.2270] | [0.0187, 0.0560] | [0.0110, 0.0422] |
| | Mistral 7B | [0.1183, 0.1920] | [0.4775, 0.5797] | [0.0190, 0.0569] | [0.0190, 0.0569] |
| | Qwen 7B | [0.1407, 0.2238] | [0.1269, 0.2071] | [0.0105, 0.0438] | [0.0105, 0.0438] |
| | QWQ | [0.8611, 0.9236] | [0.6940, 0.7833] | [0.6770, 0.7680] | [0.6375, 0.7320] |
| | GPT-oss | [0.9848, 0.9995] | [0.8709, 0.9310] | [0.8679, 0.9287] | [0.8618, 0.9240] |
| | GPT 5 mini | [0.9897, 1.0000] | [0.9650, 0.9925] | [0.9650, 0.9925] | [0.9614, 0.9908] |
| CoT pseudocode | Llama 8B | [0.3340, 0.4327] | [0.0227, 0.0627] | [0.0028, 0.0236] | [0.0028, 0.0236] |
| | Llama 70B | [0.2794, 0.3746] | [0.1139, 0.1860] | [0.0227, 0.0627] | [0.0148, 0.0492] |
| | Mistral 7B | [0.0953, 0.1641] | [0.6001, 0.6984] | [0.0277, 0.0714] | [0.0277, 0.0714] |

| Prompt | Model | Feasibility | Assignment | Matching | Optimality |
|--------|-------|-------------|------------|----------|------------|
| | Qwen 7B | [0.1327, 0.2094] | [0.1229, 0.1975] | [0.0131, 0.0463] | [0.0112, 0.0428] |
| | QWQ | [0.8011, 0.8757] | [0.6011, 0.6982] | [0.5845, 0.6825] | [0.5485, 0.6483] |
| | GPT-oss | [0.9896, 1.0000] | [0.8339, 0.9023] | [0.8339, 0.9023] | [0.8249, 0.8951] |
| | GPT 5 mini | [0.9897, 1.0000] | [0.9114, 0.9603] | [0.9114, 0.9603] | [0.9082, 0.9581] |
| CoT Python | Llama 8B | [0.5509, 0.6503] | [0.0354, 0.0822] | [0.0058, 0.0313] | [0.0058, 0.0313] |
| | Llama 70B | [0.2743, 0.3690] | [0.1778, 0.2616] | [0.0148, 0.0492] | [0.0092, 0.0386] |
| | Mistral 7B | [0.0761, 0.1396] | [0.6220, 0.7189] | [0.0235, 0.0647] | [0.0235, 0.0647] |
| | Qwen 7B | [0.1778, 0.2616] | [0.0667, 0.1260] | [0.0058, 0.0313] | [0.0005, 0.0152] |
| | QWQ | [0.7400, 0.8243] | [0.5577, 0.6576] | [0.5438, 0.6443] | [0.5078, 0.6094] |
| | GPT-oss | [0.9848, 0.9995] | [0.8317, 0.9005] | [0.8317, 0.9005] | [0.8288, 0.8981] |
| | GPT 5 mini | [0.9897, 1.0000] | [0.9724, 0.9958] | [0.9724, 0.9958] | [0.9686, 0.9942] |

### E.4  Iterative Prompting Tables

While Table 1 showed the main conclusions we can draw from iterative prompting, we present here, with Table 9-16 the exact performances for role-based prompting, but also for every other prompt to show that the conclusions drawn before stay relevant with other prompts.

### E.5  Graphs

While the Figure 7 presented the trend for every prompt, the Figure 17 and Figure 18 show the detailed results for every prompt, under both complexity dimensions, the number of students and the preference types.

Figures 16a-16e present the proportion of metrics satisfied, averaged over all instances at every iteration. For instances where the optimal matching was found before the maximal number of attempts, the value is propagated for all future iterations, and for every iteration, the bar include a number at the bottom to represent the proportion of iterations that were generated (i.e that were not previously stopped).

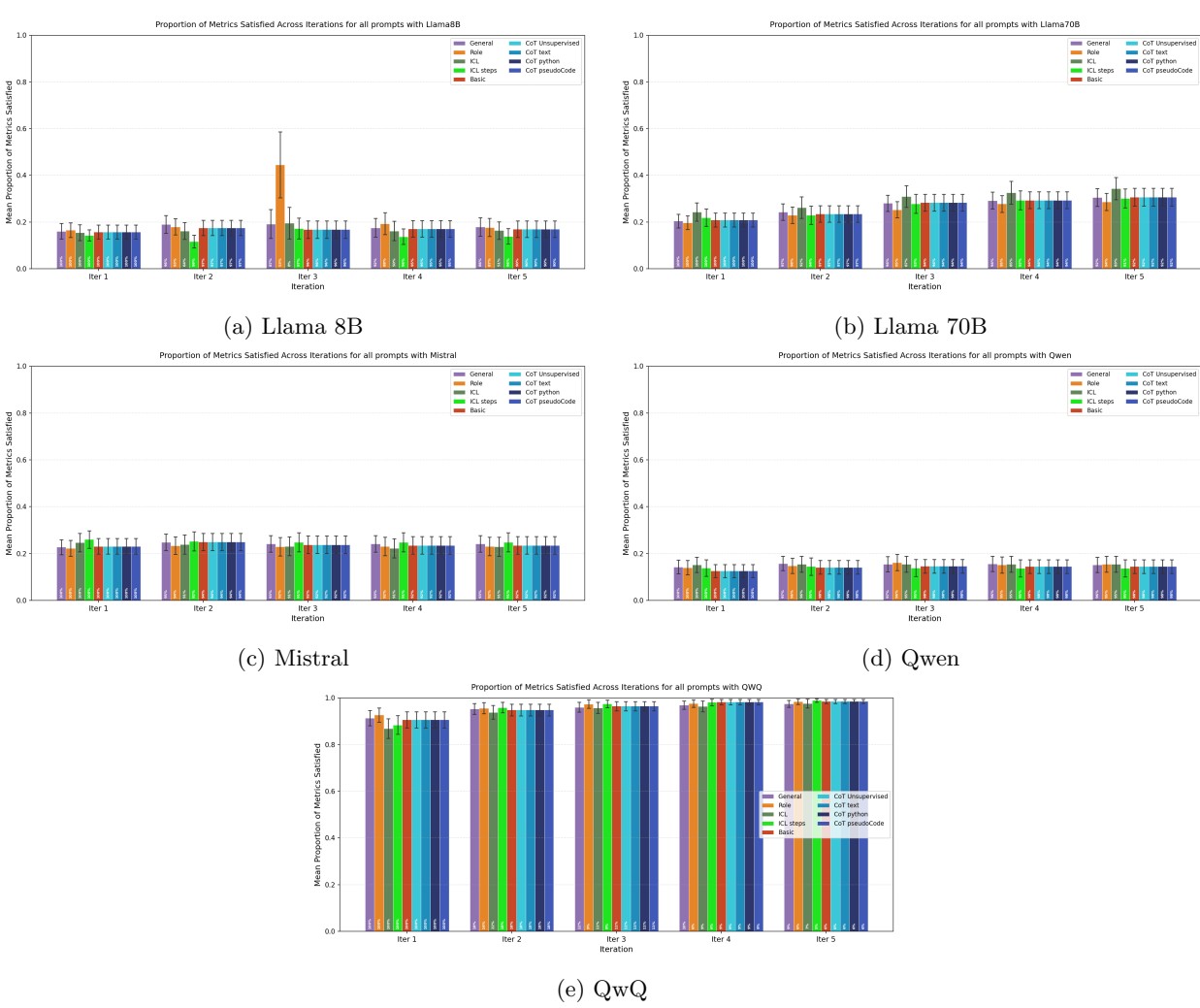

Figure 16: Metric satisfaction by iterations for iterative prompting

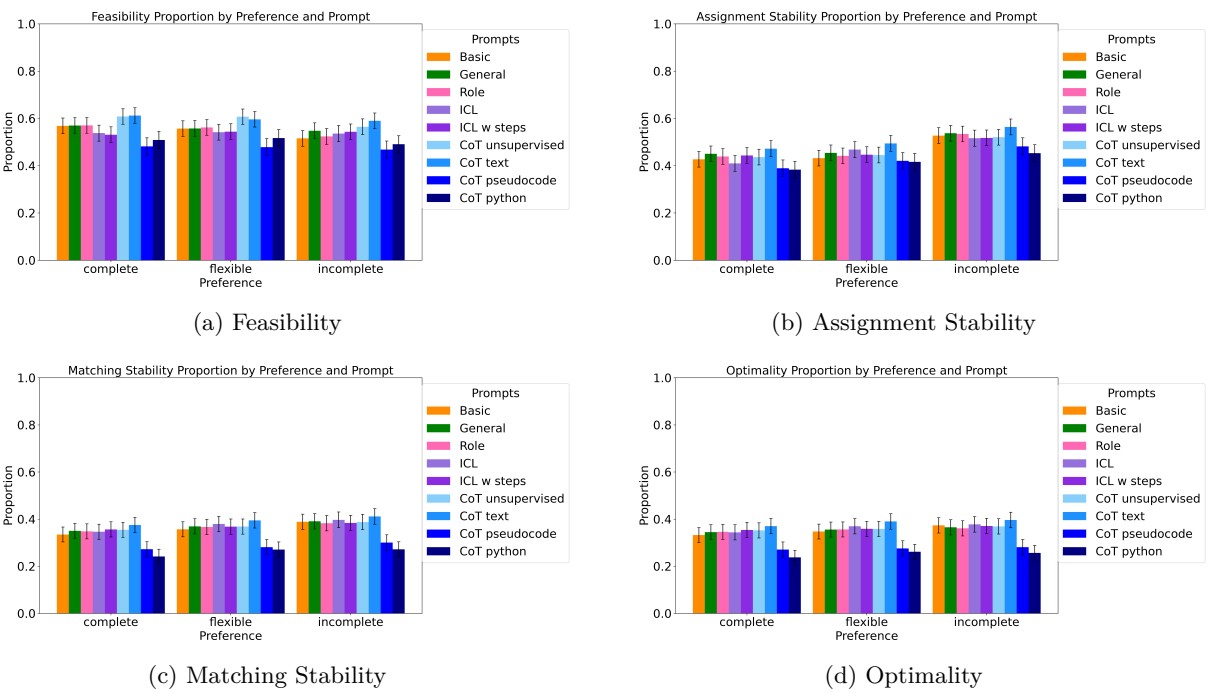

(a) Feasibility

(b) Assignment Stability

(c) Matching Stability

(d) Optimality

Figure 17: Proportion metrics by preference for all prompts.

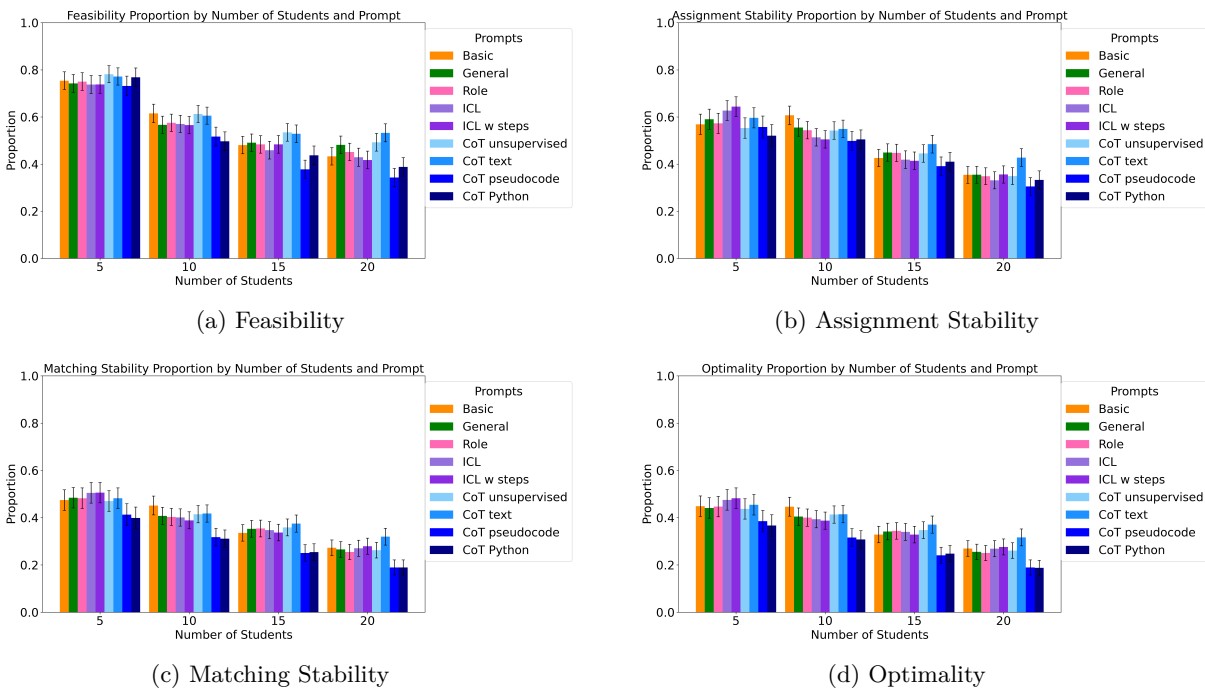

(a) Feasibility

(b) Assignment Stability

(c) Matching Stability

(d) Optimality

Figure 18: Proportion metrics by number of students for all prompts averaged over all models.

Table 5: 95% Wilson confidence interval for performance by prompts

| Type | Model | Feasibility | Assignment | Matching | Optimality |
|---|---|---|---|---|---|
| Base | Basic | [0.2341, 0.2787] | [0.1764, 0.2170] | [0.0143, 0.0290] | [0.0115, 0.0250] |
| | Role | [0.2274, 0.2715] | [0.1841, 0.2253] | [0.0155, 0.0305] | [0.0126, 0.0266] |
| | General | [0.2204, 0.2642] | [0.1947, 0.2368] | [0.0161, 0.0315] | [0.0121, 0.0259] |
| | ICL | [0.2119, 0.2563] | [0.1720, 0.2133] | [0.0242, 0.0428] | [0.0205, 0.0379] |
| | ICL w steps | [0.2060, 0.2491] | [0.1854, 0.2270] | [0.0216, 0.0390] | [0.0186, 0.0351] |
| | CoT unsup. | [0.2893, 0.3368] | [0.1660, 0.2058] | [0.0116, 0.0252] | [0.0094, 0.0219] |
| | CoT text | [0.2933, 0.3416] | [0.2049, 0.2482] | [0.0165, 0.0323] | [0.0142, 0.0291] |
| | CoT pseudo | [0.2298, 0.2743] | [0.2238, 0.2679] | [0.0214, 0.0387] | [0.0184, 0.0347] |
| | CoT python | [0.2889, 0.3364] | [0.2338, 0.2784] | [0.0167, 0.0323] | [0.0127, 0.0267] |
| Reasoning | Basic | [0.9448, 0.9686] | [0.8070, 0.8513] | [0.8023, 0.8470] | [0.7805, 0.8272] |
| | Role | [0.9419, 0.9663] | [0.8039, 0.8484] | [0.7982, 0.8433] | [0.7747, 0.8218] |
| | General | [0.9679, 0.9854] | [0.8086, 0.8527] | [0.8058, 0.8501] | [0.7756, 0.8227] |
| | ICL | [0.9093, 0.9403] | [0.7865, 0.8328] | [0.7827, 0.8293] | [0.7638, 0.8120] |
| | ICL w steps | [0.9373, 0.9629] | [0.7928, 0.8385] | [0.7928, 0.8385] | [0.7776, 0.8247] |
| | CoT unsup. | [0.9522, 0.9741] | [0.8162, 0.8595] | [0.8115, 0.8553] | [0.7897, 0.8356] |
| | CoT text | [0.9521, 0.9741] | [0.8562, 0.8950] | [0.8495, 0.8891] | [0.8332, 0.8747] |
| | CoT pseudo | [0.9326, 0.9591] | [0.7977, 0.8429] | [0.7920, 0.8377] | [0.7759, 0.8231] |
| | CoT python | [0.9114, 0.9420] | [0.8000, 0.8450] | [0.7952, 0.8407] | [0.7810, 0.8278] |

Table 7: 95% Wilson confidence interval for performance by preference types and models

| Preference | Model | Feasibility | Assignment Stability | Matching Stability | Optimality |
|---|---|---|---|---|---|
| Complete | Llama 8B | [0.4119, 0.4707] | [0.0316, 0.0556] | [0.0063, 0.0191] | [0.0063, 0.0191] |
| | Llama 70B | [0.2789, 0.3331] | [0.1668, 0.2129] | [0.0103, 0.0256] | [0.0103, 0.0256] |
| | Mistral 7B | [0.1341, 0.1782] | [0.4133, 0.4738] | [0.0164, 0.0355] | [0.0164, 0.0355] |
| | Qwen 7B | [0.2057, 0.2555] | [0.0365, 0.0619] | [0.0031, 0.0132] | [0.0025, 0.0119] |
| | QWQ | [0.8140, 0.8579] | [0.5741, 0.6322] | [0.5667, 0.6249] | [0.5537, 0.6122] |
| | GPT-oss | [0.9934, 0.9995] | [0.8058, 0.8501] | [0.8058, 0.8501] | [0.8001, 0.8450] |
| | GPT 5 mini | [0.9965, 1.0000] | [0.9308, 0.9576] | [0.9308, 0.9576] | [0.9268, 0.9544] |
| Flexible | Llama 8B | [0.3579, 0.4153] | [0.0361, 0.0613] | [0.0069, 0.0201] | [0.0025, 0.0118] |
| | Llama 70B | [0.3159, 0.3717] | [0.1771, 0.2242] | [0.0297, 0.0529] | [0.0243, 0.0457] |
| | Mistral 7B | [0.1730, 0.2203] | [0.3953, 0.4542] | [0.0305, 0.0543] | [0.0297, 0.0532] |
| | Qwen 7B | [0.1737, 0.2208] | [0.0601, 0.0913] | [0.0098, 0.0248] | [0.0083, 0.0226] |
| | QWQ | [0.8167, 0.8601] | [0.6120, 0.6687] | [0.6009, 0.6580] | [0.5705, 0.6284] |
| | GPT-oss | [0.9949, 0.9998] | [0.8494, 0.8890] | [0.8494, 0.8890] | [0.8436, 0.8839] |
| | GPT 5 mini | [0.9965, 1.0000] | [0.9647, 0.9832] | [0.9647, 0.9832] | [0.9512, 0.9734] |
| Incomplete | Llama 8B | [0.3197, 0.3758] | [0.0505, 0.0794] | [0.0147, 0.0322] | [0.0132, 0.0300] |
| | Llama 70B | [0.3185, 0.3745] | [0.1361, 0.1789] | [0.0236, 0.0447] | [0.0049, 0.0165] |
| | Mistral 7B | [0.0610, 0.0929] | [0.6711, 0.7262] | [0.0310, 0.0552] | [0.0310, 0.0552] |
| | Qwen 7B | [0.1397, 0.1834] | [0.1898, 0.2386] | [0.0211, 0.0416] | [0.0195, 0.0394] |
| | QWQ | [0.8579, 0.8965] | [0.6666, 0.7209] | [0.6482, 0.7033] | [0.5554, 0.6135] |
| | GPT-oss | [0.9934, 0.9995] | [0.9011, 0.9334] | [0.9001, 0.9326] | [0.8952, 0.9285] |
| | GPT 5 mini | [0.9949, 0.9998] | [0.9701, 0.9868] | [0.9690, 0.9861] | [0.9679, 0.9854] |

Table 8: 95% Wilson confidence interval for performance by number of students and models

| Students | Model | Feasibility | Assignment Stability | Matching Stability | Optimality |
|---|---|---|---|---|---|
| 5 | Llama 8B | [0.6122, 0.6854] | [0.1543, 0.2137] | [0.0577, 0.0986] | [0.0457, 0.0830] |
| | Llama 70B | [0.8826, 0.9274] | [0.1414, 0.1989] | [0.1158, 0.1693] | [0.0699, 0.1140] |
| | Mistral 7B | [0.3181, 0.3916] | [0.5210, 0.5973] | [0.1061, 0.1579] | [0.1047, 0.1563] |
| | Qwen 7B | [0.3899, 0.4659] | [0.2491, 0.3183] | [0.0617, 0.1037] | [0.0550, 0.0951] |
| | QWQ | [0.9941, 1.0000] | [0.9699, 0.9905] | [0.9699, 0.9905] | [0.8158, 0.8714] |
| | GPT-oss | [0.9941, 1.0000] | [0.9888, 0.9992] | [0.9888, 0.9992] | [0.9888, 0.9992] |
| | GPT 5 mini | [0.9941, 1.0000] | [0.9941, 1.0000] | [0.9941, 1.0000] | [0.9941, 1.0000] |
| 10 | Llama 8B | [0.3284, 0.3952] | [0.0416, 0.0738] | [0.0000, 0.0048] | [0.0000, 0.0048] |
| | Llama 70B | [0.3867, 0.4513] | [0.0809, 0.1201] | [0.0031, 0.0146] | [0.0031, 0.0146] |
| | Mistral 7B | [0.0610, 0.0962] | [0.6943, 0.7530] | [0.0156, 0.0360] | [0.0156, 0.0360] |
| | Qwen 7B | [0.1961, 0.2507] | [0.1066, 0.1503] | [0.0017, 0.0115] | [0.0017, 0.0115] |
| | QWQ | [0.9794, 0.9939] | [0.8388, 0.8840] | [0.8364, 0.8819] | [0.8185, 0.8662] |
| | GPT-oss | [0.9957, 1.0000] | [0.8958, 0.9323] | [0.8958, 0.9323] | [0.8933, 0.9303] |
| | GPT 5 mini | [0.9957, 1.0000] | [0.9824, 0.9954] | [0.9824, 0.9954] | [0.9795, 0.9939] |
| 15 | Llama 8B | [0.2691, 0.3293] | [0.0006, 0.0082] | [0.0000, 0.0043] | [0.0000, 0.0043] |
| | Llama 70B | [0.1128, 0.1575] | [0.1632, 0.2144] | [0.0002, 0.0063] | [0.0002, 0.0063] |
| | Mistral 7B | [0.0575, 0.0923] | [0.4224, 0.4887] | [0.0047, 0.0182] | [0.0047, 0.0182] |
| | Qwen 7B | [0.0534, 0.0869] | [0.0278, 0.0536] | [0.0000, 0.0044] | [0.0000, 0.0044] |
| | QWQ | [0.8441, 0.8887] | [0.5474, 0.6122] | [0.5235, 0.5888] | [0.4829, 0.5486] |
| | GPT-oss | [0.9919, 0.9994] | [0.8605, 0.9027] | [0.8605, 0.9027] | [0.8533, 0.8965] |
| | GPT 5 mini | [0.9957, 1.0000] | [0.9616, 0.9827] | [0.9616, 0.9827] | [0.9471, 0.9726] |
| 20 | Llama 8B | [0.2992, 0.3613] | [0.0000, 0.0044] | [0.0000, 0.0044] | [0.0000, 0.0044] |
| | Llama 70B | [0.0155, 0.0358] | [0.2402, 0.2983] | [0.0000, 0.0043] | [0.0000, 0.0043] |
| | Mistral 7B | [0.0964, 0.1416] | [0.3036, 0.3699] | [0.0000, 0.0049] | [0.0000, 0.0049] |
| | Qwen 7B | [0.1015, 0.1457] | [0.0279, 0.0543] | [0.0000, 0.0045] | [0.0000, 0.0045] |
| | QWQ | [0.5497, 0.6153] | [0.2115, 0.2683] | [0.1927, 0.2478] | [0.1839, 0.2382] |
| | GPT-oss | [0.9901, 0.9988] | [0.6981, 0.7565] | [0.6969, 0.7555] | [0.6865, 0.7457] |
| | GPT 5 mini | [0.9937, 0.9998] | [0.8920, 0.9292] | [0.8907, 0.9282] | [0.8846, 0.9231] |

Table 9: Iterative prompting results with Basic prompting on an instance with 5-10 students. For each model, we have the metrics for no iterative prompting and iterative prompting with the last and best attempt, respectively.

| Model | Iterative prompting | Feasibility | Assignment | Matching | Optimality |
|-------|--------------------|-------------|------------|----------|------------|
| Llama8B | No | 0.4561 | 0.1111 | 0.0351 | 0.0292 |
| Llama8B | Last | 0.4795 | 0.0936 | 0.0585 | 0.0468 |
| Llama8B | Best | 0.6140 | 0.1228 | 0.0702 | 0.0468 |
| Llama70B | No | 0.6491 | 0.1053 | 0.0526 | 0.0292 |
| Llama70B | Last | 0.8889 | 0.1287 | 0.1170 | 0.0819 |
| Llama70B | Best | 0.8947 | 0.1520 | 0.1345 | 0.0819 |
| Mistral 7B | No | 0.1988 | 0.6082 | 0.0585 | 0.0585 |
| Mistral 7B | Last | 0.1637 | 0.6257 | 0.0760 | 0.0760 |
| Mistral 7B | Best | 0.1871 | 0.6901 | 0.0760 | 0.0760 |
| Qwen 7B | No | 0.2690 | 0.1930 | 0.0234 | 0.0234 |
| Qwen 7B | Last | 0.4620 | 0.0643 | 0.0351 | 0.0234 |
| Qwen 7B | Best | 0.5380 | 0.1520 | 0.0351 | 0.0234 |
| QwQ 32B | No | 0.9883 | 0.9123 | 0.9064 | 0.8363 |
| QwQ 32B | Last | 1.0000 | 0.9942 | 0.9942 | 0.9357 |
| QwQ 32B | Best | 1.0000 | 0.9942 | 0.9942 | 0.9357 |

Table 10: Iterative prompting with Role prompting for instance with 5-10 students. For each model, we have the metrics for no iterative prompting and iterative prompting with the last and best attempt

| Model | Iterative prompting | Feasibility | Assignment | Matching | Optimality |
|-------|--------------------|-------------|------------|----------|------------|
| Llama 8B | No | 0.4678 | 0.1170 | 0.0409 | 0.0351 |
| Llama 8B | Last | 0.4386 | 0.1287 | 0.0760 | 0.0643 |
| Llama 8B | Best | 0.6082 | 0.1579 | 0.0877 | 0.0643 |
| Llama 70B | No | 0.6374 | 0.0877 | 0.0468 | 0.0234 |
| Llama 70B | Last | 0.8538 | 0.1170 | 0.1111 | 0.0819 |
| Llama 70B | Best | 0.8889 | 0.1404 | 0.1404 | 0.0819 |
| Mistral 7B | No | 0.2164 | 0.5556 | 0.0585 | 0.0585 |
| Mistral 7B | Last | 0.1930 | 0.5673 | 0.0819 | 0.0819 |
| Mistral 7B | Best | 0.2281 | 0.6140 | 0.0819 | 0.0819 |
| Qwen 7B | No | 0.2398 | 0.2573 | 0.0351 | 0.0351 |
| Qwen 7B | Last | 0.4503 | 0.0760 | 0.0526 | 0.0468 |
| Qwen 7B | Best | 0.5556 | 0.1930 | 0.0585 | 0.0468 |
| QwQ 32B | No | 0.9942 | 0.9064 | 0.9006 | 0.8304 |
| QwQ 32B | Last | 1.0000 | 0.9942 | 0.9942 | 0.9532 |
| QwQ 32B | Best | 1.0000 | 0.9883 | 0.9883 | 0.9415 |

Table 11: Iterative prompting results with ICL prompting on an instance with 5-10 students. For each model, we have the metrics for no iterative prompting and iterative prompting with the last and best attempt, respectively.

| Model | Iterative prompting | Feasibility | Assignment | Matching | Optimality |
|---|---|---|---|---|---|
| Llama8B | No | 0.3041 | 0.1696 | 0.0643 | 0.0585 |
| Llama8B | Last | 0.3801 | 0.1111 | 0.0819 | 0.0702 |
| Llama8B | Best | 0.4561 | 0.1228 | 0.0819 | 0.0702 |
| Llama70B | No | 0.6374 | 0.1754 | 0.0819 | 0.0643 |
| Llama70B | Last | 0.8129 | 0.1345 | 0.1287 | 0.1228 |
| Llama70B | Best | 0.8596 | 0.1871 | 0.1579 | 0.1228 |
| Mistral 7B | No | 0.2000 | 0.6667 | 0.1030 | 0.0970 |
| Mistral 7B | Last | 0.1988 | 0.6140 | 0.0936 | 0.0936 |
| Mistral 7B | Best | 0.2222 | 0.6608 | 0.0994 | 0.0936 |
| Qwen 7B | No | 0.3392 | 0.1228 | 0.0409 | 0.0351 |
| Qwen 7B | Last | 0.4327 | 0.0936 | 0.0526 | 0.0351 |
| Qwen 7B | Best | 0.4620 | 0.0994 | 0.0526 | 0.0351 |
| QwQ 32B | No | 0.9825 | 0.8889 | 0.8830 | 0.8246 |
| QwQ 32B | Last | 0.9942 | 0.9825 | 0.9766 | 0.9591 |
| QwQ 32B | Best | 0.9942 | 0.9883 | 0.9825 | 0.9591 |

Table 12: Iterative prompting results with ICL with steps prompting on an instance with 5-10 students. For each model, we have the metrics for no iterative prompting and iterative prompting with the last and best attempt, respectively.

| Model | Iterative prompting | Feasibility | Assignment | Matching | Optimality |
|---|---|---|---|---|---|
| Llama8B | No | 0.5088 | 0.0409 | 0.0117 | 0.0117 |
| Llama8B | Last | 0.3684 | 0.0994 | 0.0526 | 0.0351 |
| Llama8B | Best | 0.5731 | 0.1111 | 0.0526 | 0.0351 |
| Llama70B | No | 0.5205 | 0.2164 | 0.0936 | 0.0702 |
| Llama70B | Last | 0.8363 | 0.1579 | 0.1520 | 0.1111 |
| Llama70B | Best | 0.8713 | 0.1871 | 0.1637 | 0.1111 |
| Mistral 7B | No | 0.1754 | 0.6959 | 0.0877 | 0.0819 |
| Mistral 7B | Last | 0.2047 | 0.6023 | 0.0994 | 0.0936 |
| Mistral 7B | Best | 0.2047 | 0.6784 | 0.0994 | 0.0936 |
| Qwen 7B | No | 0.2749 | 0.1754 | 0.0526 | 0.0526 |
| Qwen 7B | Last | 0.3392 | 0.0936 | 0.0643 | 0.0526 |
| Qwen 7B | Best | 0.3743 | 0.1462 | 0.0702 | 0.0526 |
| QwQ 32B | No | 0.9942 | 0.8830 | 0.8830 | 0.8187 |
| QwQ 32B | Last | 1.0000 | 1.0000 | 1.0000 | 0.9474 |
| QwQ 32B | Best | 1.0000 | 1.0000 | 1.0000 | 0.9474 |

Table 13: Iterative prompting results with CoT unsupervised prompting on an instance with 5-10 students. For each model, we have the metrics for no iterative prompting and iterative prompting with the last and best attempt, respectively.

| Model | Iterative prompting | Feasibility | Assignment | Matching | Optimality |
|---|---|---|---|---|---|
| Llama8B | No | 0.5088 | 0.0936 | 0.0292 | 0.0234 |
| Llama8B | Last | 0.5029 | 0.0936 | 0.0643 | 0.0526 |
| Llama8B | Best | 0.6725 | 0.1287 | 0.0760 | 0.0526 |
| Llama70B | No | 0.6550 | 0.0702 | 0.0351 | 0.0175 |
| Llama70B | Last | 0.8889 | 0.1228 | 0.1228 | 0.0702 |
| Llama70B | Best | 0.9064 | 0.1345 | 0.1287 | 0.0702 |
| Mistral 7B | No | 0.2222 | 0.6023 | 0.0526 | 0.0526 |
| Mistral 7B | Last | 0.1813 | 0.6433 | 0.0702 | 0.0702 |
| Mistral 7B | Best | 0.2164 | 0.6842 | 0.0702 | 0.0702 |
| Qwen 7B | No | 0.4094 | 0.1404 | 0.0234 | 0.0234 |
| Qwen 7B | Last | 0.4795 | 0.0643 | 0.0292 | 0.0234 |
| Qwen 7B | Best | 0.6257 | 0.1053 | 0.0292 | 0.0234 |
| QwQ 32B | No | 1.0000 | 0.9529 | 0.9529 | 0.9000 |
| QwQ 32B | Last | 1.0000 | 1.0000 | 1.0000 | 0.9532 |
| QwQ 32B | Best | 1.0000 | 1.0000 | 1.0000 | 0.9532 |

Table 14: Iterative prompting results with CoT text prompting on an instance with 5-10 students. For each model, we have the metrics for no iterative prompting and iterative prompting with the last and best attempt, respectively.

| Model | Iterative prompting | Feasibility | Assignment | Matching | Optimality |
|---|---|---|---|---|---|
| Llama8B | No | 0.6550 | 0.0526 | 0.0117 | 0.0117 |
| Llama8B | Last | 0.4503 | 0.0936 | 0.0468 | 0.0409 |
| Llama8B | Best | 0.7485 | 0.1170 | 0.0526 | 0.0409 |
| Llama70B | No | 0.6257 | 0.1345 | 0.0702 | 0.0468 |
| Llama70B | Last | 0.8070 | 0.1345 | 0.1287 | 0.0994 |
| Llama70B | Best | 0.8538 | 0.1930 | 0.1579 | 0.0994 |
| Mistral 7B | No | 0.1988 | 0.6257 | 0.0643 | 0.0643 |
| Mistral 7B | Last | 0.1988 | 0.6082 | 0.0702 | 0.0702 |
| Mistral 7B | Best | 0.2047 | 0.6374 | 0.0702 | 0.0702 |
| Qwen 7B | No | 0.2515 | 0.2573 | 0.0409 | 0.0409 |
| Qwen 7B | Last | 0.3509 | 0.1228 | 0.0468 | 0.0468 |
| Qwen 7B | Best | 0.4094 | 0.2164 | 0.0468 | 0.0468 |
| QwQ 32B | No | 1.0000 | 0.9474 | 0.9474 | 0.8947 |
| QwQ 32B | Last | 1.0000 | 1.0000 | 1.0000 | 0.9591 |
| QwQ 32B | Best | 1.0000 | 1.0000 | 1.0000 | 0.9591 |

Table 15: Iterative prompting results with CoT python prompting on an instance with 5-10 students. For each model, we have the metrics for no iterative prompting and iterative prompting with the last and best attempt, respectively.

| Model | Iterative prompting | Feasibility | Assignment | Matching | Optimality |
|---|---|---|---|---|---|
| Llama8B | No | 0.4912 | 0.1111 | 0.0292 | 0.0292 |
| Llama8B | Last | 0.4503 | 0.1170 | 0.0643 | 0.0643 |
| Llama8B | Best | 0.6199 | 0.1579 | 0.0702 | 0.0643 |
| Llama70B | No | 0.6023 | 0.1345 | 0.0585 | 0.0409 |
| Llama70B | Last | 0.7953 | 0.1404 | 0.1345 | 0.0994 |
| Llama70B | Best | 0.8304 | 0.2047 | 0.1579 | 0.0994 |
| Mistral 7B | No | 0.1696 | 0.8129 | 0.0819 | 0.0819 |
| Mistral 7B | Last | 0.1871 | 0.7368 | 0.0877 | 0.0877 |
| Mistral 7B | Best | 0.1930 | 0.8012 | 0.0877 | 0.0877 |
| Qwen 7B | No | 0.4094 | 0.1228 | 0.0292 | 0.0058 |
| Qwen 7B | Last | 0.4269 | 0.1170 | 0.0117 | 0.0117 |
| Qwen 7B | Best | 0.4737 | 0.1462 | 0.0351 | 0.0117 |
| QwQ 32B | No | 0.9824 | 0.9000 | 0.8882 | 0.8235 |
| QwQ 32B | Last | 1.0000 | 0.9942 | 0.9942 | 0.9415 |
| QwQ 32B | Best | 1.0000 | 0.9942 | 0.9942 | 0.9415 |

Table 16: Iterative prompting results with CoT pseudocode prompting on an instance with 5-10 students. For each model, we have the metrics for no iterative prompting and iterative prompting with the last and best attempt, respectively.

| Model | Iterative prompting | Feasibility | Assignment | Matching | Optimality |
|---|---|---|---|---|---|
| Llama8B | No | 0.5263 | 0.0760 | 0.0175 | 0.0175 |
| Llama8B | Last | 0.4678 | 0.1111 | 0.0760 | 0.0760 |
| Llama8B | Best | 0.6316 | 0.1637 | 0.0877 | 0.0760 |
| Llama70B | No | 0.6374 | 0.1696 | 0.0760 | 0.0526 |
| Llama70B | Last | 0.7485 | 0.1579 | 0.1520 | 0.1228 |
| Llama70B | Best | 0.8304 | 0.2164 | 0.1754 | 0.1228 |
| Mistral 7B | No | 0.1754 | 0.7661 | 0.0877 | 0.0877 |
| Mistral 7B | Last | 0.1930 | 0.6140 | 0.0877 | 0.0877 |
| Mistral 7B | Best | 0.1930 | 0.7544 | 0.0877 | 0.0877 |
| Qwen 7B | No | 0.3158 | 0.2281 | 0.0468 | 0.0468 |
| Qwen 7B | Last | 0.4094 | 0.0994 | 0.0643 | 0.0585 |
| Qwen 7B | Best | 0.4737 | 0.1871 | 0.0643 | 0.0585 |
| QwQ 32B | No | 0.9766 | 0.9357 | 0.9181 | 0.8772 |
| QwQ 32B | Last | 1.0000 | 0.9883 | 0.9883 | 0.9591 |
| QwQ 32B | Best | 1.0000 | 0.9942 | 0.9942 | 0.9591 |

Table 17: Iterative prompting results with General prompting on instances with 5-10 students. For each model, we have the metrics for no iterative prompting and iterative prompting with the last and best attempt, respectively.

| Model | Iterative prompting | Feasibility | Assignment | Matching | Optimality |
|-------|---------------------|-------------|------------|----------|------------|
| Llama8B | No | 0.3743 | 0.1754 | 0.0585 | 0.0351 |
| Llama8B | Last | 0.4561 | 0.1170 | 0.0877 | 0.0548 |
| Llama8B | Best | 0.5731 | 0.1696 | 0.0994 | 0.0585 |
| Llama70B | No | 0.6725 | 0.0702 | 0.0465 | 0.0292 |
| Llama70B | Last | 0.8889 | 0.1287 | 0.1287 | 0.0877 |
| Llama70B | Best | 0.9181 | 0.1520 | 0.1404 | 0.0877 |
| Mistral 7B | No | 0.1637 | 0.7076 | 0.0702 | 0.0702 |
| Mistral 7B | Last | 0.1520 | 0.6725 | 0.0702 | 0.0702 |
| Mistral 7B | Best | 0.1637 | 0.7076 | 0.0702 | 0.0702 |
| Qwen 7B | No | 0.3041 | 0.1930 | 0.0292 | 0.0292 |
| Qwen 7B | Last | 0.4444 | 0.0819 | 0.0468 | 0.0351 |
| Qwen 7B | Best | 0.5673 | 0.1579 | 0.0526 | 0.0351 |
| QwQ 32B | No | 0.9883 | 0.9006 | 0.8889 | 0.7836 |
| QwQ 32B | Last | 1.0000 | 0.9883 | 0.9883 | 0.9181 |
| QwQ 32B | Best | 1.0000 | 0.9883 | 0.9883 | 0.9181 |

