# OpenReview forum: "Reasoning with Preference Constraints: A Benchmark for Language Models in Many-to-One Matching Markets"
_TMLR — Decision pending for TMLR_

### Review · Reviewer_TD7v · 2026-03-15

**Summary Of Contributions:**

**Summary of the contributions**

- The authors introduce a controlled benchmark dataset for the College Admissions matching problem to evaluate LLM reasoning in matching markets.


- They propose four evaluation metrics (feasibility, assignment stability, matching stability, and student optimality) grounded in the theoretical properties of matching markets.


- They evaluate several open-source LLMs on this benchmark using these metrics.


- They analyze model performance across different problem settings (number of students, preference completeness, capacity regimes) and prompting strategies (e.g., CoT, ICL).


- They also investigate whether iterative prompting with feedback can improve model performance on this task.

**Weaknesses**

- A key limitation of the paper is that it does not evaluate the performance of leading closed-source models (e.g., ChatGPT-5.3, Claude-4.6, or Gemini-3), which could provide a stronger baseline for comparison.

- The benchmark uses randomly generated preferences, which may not reflect real-world preference structures.

**Strengths**

- The dataset allows controlled experiments by varying parameters such as student numbers and preference types. This helps isolate which factors affect model performance.
- The four metrics reflect real theoretical properties of matching markets

**Additional Comments:**

NA

**Audience:**

Yes

**Audience Explanation:**

Yes. The paper studies the ability of LLMs to solve preference-based combinatorial problems, specifically many-to-one matching markets such as the College Admissions problem. This is relevant to researchers working on LLM reasoning, algorithmic reasoning, and applications of LLMs to optimization and economic mechanisms. The proposed benchmark and evaluation metrics may also be of interest to the community.

**Claims And Evidence:**

Yes

**Claims Explanation:**

The empirical evaluation is thorough and systematically analyzes several experimental factors. However, even though I see value in the proposal of the dataset, the metrics and the experiments, **I am not fully convinced by the strength of the authors’ claim that “current LLMs struggle to reliably implement procedures such as DA.”** The models evaluated are relatively small and, in some cases, somewhat outdated, and the study does not include evaluations of stronger closed-source models. This makes it difficult to draw conclusions about the capabilities of the current state-of-the-art. Moreover, the results themselves suggest that some models perform very well in simpler settings (e.g., GPT-oss achieves near-perfect optimality for instances with 5 students in Figure 5), which indicates that the difficulty may stem more from scaling the instance size rather than a fundamental inability to perform the reasoning required. This raises the possibility that the observed failures may be influenced by experimental constraints (e.g., output length, task formulation, or scaling of the instance) rather than inherent reasoning limitations, a concern that has been raised in similar debates about reasoning benchmarks (for example, in “The Illusion of Thinking: Understanding the Strengths and Limitations of Reasoning Models via the Lens of Problem Complexity” (Shojaee et al., 2025), where strong performance degradation on structured reasoning tasks was later debated as potentially reflecting evaluation design choices rather than fundamental reasoning deficits).

**Requested Changes:**

**Requested Changes**

1. Evaluation with stronger models.

If possible, I would encourage the authors to include experiments with some leading closed-source models (e.g., recent ChatGPT, Claude, or Gemini variants). Since the paper makes claims about the limitations of “current LLMs,” including stronger state-of-the-art systems would help determine whether the observed limitations are due to the models evaluated or are more fundamental.

2. Clarify the relationship between stability notions.

It would help readers if the paper made it more explicit early on that matching stability = feasibility + assignment stability. While this relationship is implicit in the definitions, stating it clearly would improve readability and avoid confusion when interpreting the evaluation metrics.

3. Terminology on page 2.

I would avoid referring to the College Admission problem as a “multi-agent system problem.” The term “agent” is strongly associated with reinforcement learning or autonomous agents, which could lead to misunderstandings about the intended meaning.

4. Notation for acceptable pairs (page 3).

The definition of acceptable pairs could be simplified. Since it will never be the case that $\emptyset \succ_s \emptyset$, it seems unnecessary to write $E \subseteq S \times (C \cup \{\emptyset\})$ in that definition. Restricting the notation to pairs involving actual colleges would make the definition clearer.

5. Commentary on Figure 5a.

If there are insights from the experiments, it would be useful to comment on the observation that Llama 8B appears to outperform Llama 70B for instances with a larger number of students.

6. Clarification regarding Figure 7a.

I am a bit confused about Figure 7a. If optimality requires feasibility, how can optimality appear higher than feasibility for some lines (e.g., the CoT pseudocode line)?

7. Table 1 typos and potential inconsistency.

- For QwQ, “Non” should likely be “No.”
- In Table 1, the “Best” row for QwQ appears to have slightly worse performance than the “Last” row. Could these rows have been accidentally swapped?

8. Tone of claims (page 12 and conclusion).

I would recommend softening statements such as “current LLMs struggle to reliably implement procedures such as DA “ or “solving a two sided many to one matching problem remains highly challenging for LLMs”  unless experiments with stronger closed-source models are included. The current experiments are limited to a specific set of models, some of which are relatively small.

9. Discussion of tool use.

It may also be worth discussing whether stronger models could solve larger instances by generating or executing code (e.g., implementing the DA algorithm directly) rather than reasoning purely in text. This could change the interpretation of the results for larger problem instances.

---

> ### Author Response · Authors · 2026-04-11
> **Addressing Requested Changes: stronger models and stability notions**
>
> Firstly, we would like to thank the reviewer for their constructive comments and valuable suggestions. Below, we address each of the requested changes, explaining either why certain suggestions could not be implemented or presenting the additional results and conclusions obtained, along with a description of how the manuscript has been revised accordingly.
>
> 1. Evaluation of stronger models and Claims
>
> We thank the reviewers for highlighting the importance of evaluating recent, high-performing closed-source models. We agree that including  closed-source models could strengthen our claims and help determine whether the observed limitations are fundamental to large language models. However, such models are not easily accessible and present important challenges for reproducibility and fairness of comparison.
>
> First, our primary objective in this paper is to ensure accessibility and reproducibility. Closed-source models often restrict control over key parameters. For example, we enforce deterministic decoding (temperature = 0) in our experiments, which is not consistently supported by reasoning-focused models such as GPT-o3. As a result, outputs would not be generated under equivalent configurations across models, making comparisons inherently unfair. One possible workaround would be to run each instance multiple times to account for stochasticity, but this would substantially increase the number of required experiments.
>
> Second, cost and accessibility constitute significant barriers, which ideally should be minimized in academic research. Evaluating all 369 instances across all prompt strategies, particularly given the rapidly increasing token length of larger problem instances and the potential need for repeated runs to compensate for non-deterministic settings, would incur substantial cost.
> To address these limitations, we designed our benchmark to be fully extensible and reproducible: all problem instances are publicly available, and additional instances of arbitrary size can be generated using our code. This enables future work to systematically evaluate stronger or closed-source models within the same framework.
>
> Accordingly, we have revised the manuscript to clarify the scope of our claims. In particular, we explicitly state that our conclusions apply to the selected open-weight models evaluated in this study, and we avoid generalizing to all LLMs. We also highlight the evaluation of closed-source frontier models as an important direction for future work (Section 4), and we have updated Section 1 and related passages to reflect this distinction.
>
> Following the other reviewer’s suggestion, we also introduced a new prompt, “General,” which removes any explicit reference to the DA algorithm. This prompt does not lead to improved performance and is never the best-performing option.
> While conducting these additional experiments, we re-evaluated all models on the original prompts to ensure consistency and observed that GPT-oss did not reproduce its previous results. This model is very large and could not be stored. There is also no versioning and only one snapshot on the openAI website, which shows no date. Consequently, we reran all experiments involving this model and updated our conclusions.
>
> Overall, feasibility remains nearly perfect across all instance sizes. For other, less straightforward metrics, performance declines as instance size increases, although less sharply than previously observed. While overall performance has improved, instances with 20 students remain relatively small for matching problems, yet already exhibit increasing complexity. Therefore, our overall conclusions remain valid, but we put more emphasis on the fact that it seems like some models could understand the required notions at a small scale, but that scaling makes it very fast challenging. Our results are still relevant to show that using such models in combinatorial optimization for heuristic generation could be problematic, when we see they are not robust for a relatively simple and small problem.
>
> 2. Stability notions
>
> We formally define these notions in the Problem Statement section. To improve clarity, we have explicitly distinguished between the two notions of stability immediately following their definitions in Section 3.1. In particular, after introducing matching stability, we add the following clarification: “Matching stability requires both feasibility and assignment stability. In contrast, assignment stability does not require feasibility and is therefore a weaker notion.” We hope this addition clarifies the distinction between the two concepts from the beginning.

---

> ### Author Response · Authors · 2026-04-11
> **Requested Changes: Terminology and Results**
>
> 3. “Agents” terminology
>
> We agree with the reviewer that the use of the term agent (or multi-agent) may be misleading given its established meaning in the literature. In our context, it was intended to refer collectively to students and colleges; however, we recognize that this terminology may introduce confusion. Accordingly, we have removed references to agents and instead adopted more precise and neutral terminology, specifically as participants, which is defined in Section 1.
>
> 4. Acceptable pairs definition
>
> We thank the reviewer for mentioning the lack of clarity in the definition. Because it is important that a pair (s, \emptyset) is considered acceptable, we did not change the definition of what is an acceptable pair, as it also follows prior works [1]. To improve clarity and avoid the confusion mentioned by the reviewer, we instead changed the verbal explanation, instead describing what would be a pair that is not acceptable. The new definition has been revised in section 3.1.
>
> [1] Bobbio, F., Carvalho, M., Lodi, A., Rios, I., & Torrico, A. (2025). Capacity planning in stable matching. Operations Research.
>
> 5. Figures and tables
>
> Figure 5:
> We observe an unexpected trend in Figure 5: Llama 8B outperforms Llama 70B as the problem size increases. This difference is statistically significant as observed with the 95% confidence interval at 20 students which have been added for all Figures, along with the corresponding tables in the appendix D.3.
>
> This suggests that scaling model size does not necessarily translate into improved performance on this task. While we do not have direct insight that would allow us to precisely diagnose this behavior, we agree that this is an interesting and non-trivial finding, and it should be mentioned.
>
> It is also important to note that the smaller model only outperforms the larger one in terms of feasibility, and primarily on larger instances. One possible explanation is that smaller models may implicitly prioritize simpler or more explicit constraints (such as feasibility), while failing to consistently account for more complex criteria such as stability and optimality. In contrast, larger models may attempt to balance multiple constraints simultaneously, leading to different trade-offs that become more pronounced as problem complexity increases. This potential explanation, along with the surprising results, have been added under RQ1 of Section 4.
>
> Figure 6 and 7:
> ​​We thank the reviewer for pointing out the confusion caused by incorrect captions.
> * Figure 6 corresponds to bar plots over preference types and reports the proportion of matchings satisfying each metric. Its corrected caption will be:
>  “Proportion metrics by preference type and model.”
>  The term “normalized” will be removed.
>
> * Figure 7 presents the spider plots. Its corrected caption will be:
>  “Normalized performance by model type and prompting strategy.”
>
>
> The normalization is performed independently along each metric axis to improve visibility, as some metrics (e.g., optimality) have very small absolute values. As a result, this figure is intended to compare relative performance across prompting strategies within each metric, and not across metrics. All changes have been made for the respective Figures of Section 4.
>
> Table 1:
> We thank the reviewer for identifying the typo, which will be corrected.
> Regarding the apparent inconsistency between the “Best” and “Last” rows, we reran all models while ensuring that each run recorded the first matching (i.e., without iterative prompting), as well as the best and the last outcomes across a maximum of five attempts per model. Although we enforced deterministic decoding (temperature = 0), minor sources of variability (e.g., tokenization effects) may still introduce small differences across runs, which explains the few minor changes in the reported values.

---

> > ### Comment · Reviewer_TD7v · 2026-04-12
> > **Response to Authors: Remaining Concerns on Reproducibility, Cost, and Methodology**
> >
> > I thank the authors for their detailed responses and the effort put into revising the manuscript. Several of the changes are welcome improvements. Below I address the remaining points, including some new concerns raised by the authors' response.
> >
> >
> > - **On the cost and reproducibility arguments for excluding closed-source models.**
> >
> >  I appreciate the authors' transparency on this point, but I find the arguments less convincing than they may appear. Current API pricing for frontier reasoning models is very low. The authors' benchmark consists of 369 relatively short instances (5–20 students) across 9 prompt strategies. A rough estimate puts the total cost of evaluating a single closed-source model at well under $400.
> >
> >
> > The reproducibility argument regarding temperature=0 not being universally supported by reasoning models is valid in principle, but the standard practice in the field is to either set temperature=0 where supported or run a small number of seeds (3–5) and report means with confidence intervals, which the authors already do for their open-weight experiments. This is a solvable problem, not a blocking one.
> >
> >
> > The expectation in the community is that benchmarking papers include at least some frontier closed-source models, particularly when the paper's central claims concern the limitations of "current LLMs."
> >
> >
> > While the revised framing (restricting conclusions to the evaluated open-weight models) is an improvement, I believe the paper's contribution would be substantially strengthened by including even one or two frontier models.
> >
> >
> > - **On GPT-oss reproducibility — a significant concern.**
> >
> > The authors write:
> >
> >
> >
> > _"We re-evaluated all models on the original prompts to ensure consistency and observed that GPT-oss did not reproduce its previous results. This model is very large and could not be stored. There is also no versioning and only one snapshot on the openAI website, which shows no date."_
> >
> >
> > I find this paragraph concerning, and I would ask the authors to clarify several points.
> >
> >
> > GPT-oss-120b is an open-weight model released, with weights publicly hosted on Hugging Face (openai/gpt-oss-120b). It is not a closed-source API model and the Hugging Face repository has standard git-based version history with commit hashes that can be pinned for exact reproducibility.
> >
> >
> > The claim that there is no versioning and only one snapshot on the OpenAI website, which shows no date appears inaccurate, as the Hugging Face repository both have dates, and HF commit pinning is the standard mechanism for versioning open-weight models (https://huggingface.co/openai/gpt-oss-120b/commits/main).
> >
> >
> > This raises important methodological questions:
> >
> > - How was GPT-oss served in the original and re-run experiments? Was it self-hosted, or was a third-party inference API used? If the latter, this would explain the non-reproducibility (the provider may have changed quantization or serving software), but it also means the authors were not controlling for a key experimental variable.
> >
> > - What specifically changed in the reproduced results? The authors mention they "updated our conclusions" but do not clearly describe whether GPT-oss performance improved, degraded, or shifted in a more nuanced way.
> >
> > - Some community benchmarks have reported performance inconsistency with GPT-oss depending on deployment configuration and quantization level. Could differences in inference setup (e.g., different quantization) explain the discrepancy? If that is the case, why has the setup changed from the original experiments to the current ones?
> >
> >
> > For a benchmarking paper, exact reproducibility of the experimental setup is essential. I would ask the authors to provide full details of their inference configuration for GPT-oss (and ideally all models): framework, quantization, hardware, commit hash or model version from huggingface if they are worried about multiple versions of GPT OSS, and any system prompts or special formatting used.
> >
> >
> > - **On tool use / code generation (not addressed).**
> >
> > I note that this point from my original review was not addressed in the response. The question of whether frontier models could solve these instances by generating and executing DA code, rather than reasoning purely in natural language, remains relevant to the interpretation of the results. Even a brief discussion acknowledging this as an alternative paradigm would strengthen the paper's framing. This is particularly pertinent given the authors' own suggestion that LLMs might serve as tools to assist practitioners rather than as direct solvers.

---

> > > ### Author Response · Authors · 2026-05-19
> > > **Remaining Concerns**
> > >
> > > **Cost Closed Source Model**
> > >
> > > We agree with the reviewer that the price per one million tokens can be relatively low for some closed-source models. We also estimated that it would cost approximately $500 USD for most openAI models. However, several concerns remain.
> > >
> > > 1. Regarding reproducibility, as noted by the reviewer, running multiple seeds for the same instance could mitigate the issue of not being able to set the temperature to 0. However, in our dataset, the 3 seeds used for each configuration correspond to different instances, since the preferences are generated randomly and therefore vary across seeds. This choice was made intentionally to evaluate robustness across distinct problem instances. Consequently, to ensure a truly fair comparison, we would need to run each of the 369 instances approximately 3–5 times. This would increase the overall cost by the same factor, resulting in a substantially larger expense.
> > > 2. Although this amount may appear reasonable in some contexts, it is nevertheless not necessarily inexpensive, particularly when accounting for currency conversion rates and the practical difficulties associated with obtaining funding for API usage.
> > >
> > > Therefore, we could find it difficult to afford to run such models, but we do understand the importance of including such models to contextualize our findings and determine whether closed-source models yield fundamentally different conclusions. As a compromise, we are wondering whether using a closed-source model that is relatively cheaper would be satisfactory, like GPT 5-mini or Grok 4.1 Fast. This would allow us to maintain the scope of the paper on an accessible benchmarking pipeline.
> > >
> > > **GPT oss**
> > >
> > > We agree that some points regarding the generation of responses with GPT-oss 120B require clarification. We also agree with the reviewer that the model is publicly available and that its Hugging Face page indicates that the weights have remained unchanged since their release last August.
> > >
> > >
> > > We ran the model locally (self-hosted), but we had to recreate the environment and re-download the model when conducting the additional experiments. At that point, we observed the changes in the results. As the reviewer correctly pointed out, GPT-oss is highly sensitive to implementation details. After investigating the source of the discrepancies as much as possible, we found that the earlier results had been obtained using older versions of certain packages (notably Triton), which caused the MXFP4 quantization to fall back to BF16. Reverting to this setup explains more than half of the observed performance gap, although it does not exactly reproduce the original results, likely due to additional versioning differences.
> > >
> > >
> > > Reproducibility is an important objective of this work. To strengthen this aspect, and following the reviewer’s helpful suggestion, we added a dedicated appendix section detailing the inference configuration.
> > >
> > >
> > > Finally, because the updated results are just generally better across all metrics and prompts, we did not explicitly distinguish between the previous and revised results throughout the paper. However, we moderated our claims to suggest that better models might at least have some understanding of the problem and their properties, even if they are still unable to solve it at the scale at which it is typically addressed.
> > >
> > >
> > > **Tools**
> > > The comment regarding the use of tools has now been incorporated into the discussion (Section 5). We agree that access to tools would likely improve model performance, particularly for prompts such as CoT Python, which include an implementation of the DA algorithm specifically written for the structure of the instances.
> > >
> > >
> > > Although this would likely lead to improved performance, the goal of this paper is primarily to study the models’ understanding of core notions around preferences, such as stability. In that sense, we believe the results still serve as a caution against relying on LLMs, with or without tools, to solve more complex problems in combinatorial optimization, including proposing heuristic approaches for NP-hard problems, as mentioned in the discussion.

---

### Review · Reviewer_qZyU · 2026-03-16

**Summary Of Contributions:**

This paper introduces a benchmark of 369 synthetic instances of the College Admission Problem—a many-to-one matching market—to evaluate how LLMs reason over preference-based constraints. The benchmark systematically varies the number of students, student-college ratios, preference completeness, and capacity regimes. Six open-weight models, including Llama 3 (8B/70B), Mistral 7B, Qwen2 7B, QwQ 32B, and GPT-oss 120B, are evaluated across eight distinct prompting strategies such as Role, ICL, and various Chain-of-Thought (CoT) formats. The authors propose four hierarchical metrics: feasibility, assignment stability, matching stability, and student-optimality. Key findings indicate that reasoning-trained models like QwQ and GPT-oss significantly outperform base models, though performance degrades as instance size increases and no single prompting strategy emerges as a universal winner.

**Audience:**

Yes

**Audience Explanation:**

Researchers at the intersection of LLM reasoning, combinatorial optimization, and game theory will find these findings relevant. The observation that even specialized reasoning models struggle with polynomial-time solvable problems at small scales is highly informative for those developing LLM-based agents for market design or operations research. This systematic comparison provides a broader understanding of how to elicit structured reasoning from LLMs in formally verifiable tasks.

**Broader Impact Concerns:**

There are no significant broader impact concerns. The benchmark uses synthetic data and does not propose deploying LLMs as actual matching market solvers. In fact, the results serve as a caution against such deployments by highlighting existing technical limitations.

**Claims And Evidence:**

Yes

**Claims Explanation:**

The core empirical claims are generally well-supported. The hierarchical metric structure is formally grounded, and evaluating against the ground truth of the Deferred Acceptance (DA) algorithm is a sound methodology. The superior performance of reasoning models is convincingly demonstrated; for instance, QwQ achieves approximately 0.87 feasibility compared to Llama 8B’s 0.35. Furthermore, the radar plots and detailed tables effectively evidence the non-monotonic behavior of iterative prompting and the lack of a dominant prompting strategy.

Some claims could be strengthened with more rigorous analysis. While the paper frames the benchmark as testing "reasoning with preference constraints," the prompts explicitly instruct models to "implement the student-proposing deferred acceptance algorithm." This suggests the benchmark evaluates algorithmic simulation more than general preference reasoning. Additionally, the claim that preference types have limited impact would benefit from formal statistical testing rather than the current reliance on visual inspection of the figures.

**Requested Changes:**

Inclusion of Frontier Models: The paper currently only evaluates open-weight models. Including at least one or two API-accessible closed-source frontier models, such as GPT-o3 or Claude Opus, is essential to determine if current failures reflect fundamental LLM limitations or merely capability gaps in the selected models.

Refining the Conceptual Contribution: Because models are explicitly told to use the Gale-Shapley DA algorithm, the authors must clarify if the goal is to evaluate algorithm simulation or general reasoning. The contribution should either be reframed as an evaluation of "algorithm simulation on structured inputs" or include experiments where the algorithm is not specified in the prompt, allowing models to reason from first principles.

Statistical Significance Analysis: With only three random seeds per configuration, the authors should report confidence intervals or significance tests. This is particularly important for supporting claims regarding the differences in prompting strategies and the effects of preference types.

---

> ### Author Response · Authors · 2026-04-11
> **Additional Prompt and Confidence Interval**
>
> Firstly, we would like to thank the reviewer for their constructive comments and valuable suggestions. Below, we address each of the requested changes, explaining either why certain suggestions could not be implemented or presenting the additional results and conclusions obtained, along with a description of how the manuscript has been revised accordingly.
>
> 1. Inclusion or exclusion of frontier model
>
> We thank the reviewers for highlighting the importance of evaluating recent, high-performing closed-source models. We agree that including closed-source models could strengthen our claims and help determine whether the observed limitations are fundamental to large language models. However, such models are not easily accessible and present important challenges for reproducibility and fairness of comparison.
>
> First, our primary objective in this paper is to ensure accessibility and reproducibility. Closed-source models often restrict control over key parameters. For example, we enforce deterministic decoding (temperature = 0) in our experiments, which is not consistently supported by reasoning-focused models such as GPT-o3. As a result, outputs would not be generated under equivalent configurations across models, making comparisons inherently unfair. One possible workaround would be to run each instance multiple times to account for stochasticity, but this would substantially increase the number of required experiments.
>
> Second, cost and accessibility constitute significant barriers, which ideally should be minimized in academic research. Evaluating all 369 instances across all prompt strategies, particularly given the rapidly increasing token length of larger problem instances and the potential need for repeated runs to compensate for non-deterministic settings, would incur substantial cost.
> To address these limitations, we designed our benchmark to be fully extensible and reproducible: all problem instances are publicly available, and additional instances of arbitrary size can be generated using our code. This enables future work to systematically evaluate stronger or closed-source models within the same framework.
>
> Accordingly, we have revised the manuscript to clarify the scope of our claims. In particular, we explicitly state that our conclusions apply to the selected open-weight models evaluated in this study, and we avoid generalizing to all LLMs. We also highlight the evaluation of closed-source frontier models as an important direction for future work (Section 4), and we have updated Section 1 and related passages to reflect this distinction.

---

> ### Author Response · Authors · 2026-04-11
> **Additional Prompt and Confidence Interval (2/3)**
>
> 2. Reasoning without the DA algorithm instructions
>
> The reviewer raises a very interesting conceptual point regarding the prompt. The original formulation explicitly mentioned the use of the DA algorithm, which may emphasize algorithmic simulation rather than core reasoning.
>
> First, it is worth noting that while there are multiple algorithms to solve the many-to-one matching problems. Each of them have a different objective, meaning not all the same properties are satisfied under each of them for the optimal solution. For the DA algorithm, we find the matching minimizing the sum of ranks, among all stable outcomes. Solving with another algorithm could lead to a solution that does not meet all criteria because the chosen algorithm was not optimizing for that. The DA algorithm is also very popular, known for many years, and used in school choice systems like in Chile. We expect the LLM to be aware of this algorithm.
>
> That said, the reviewer’s concern is valid, and we have introduced additional experiments to address it. Specifically, we propose a new prompt, labeled “General” in Section 3.3, which retains the definition of key properties (e.g., stability) but removes any explicit reference to the DA algorithm. This experiment strengthens our evaluation of genuine reasoning and allows us to measure the impact of explicitly mentioning the algorithm.
>
> Because the new prompts required running all models, we also made sure that our previous results stayed consistent: we also verified the consistency of our previous results. For most models, we used a fixed version selected at the beginning of the project. All results stayed consistent except for GPT-oss, which we could not keep downloaded locally due to the size. There is no versioning of the models, and the only snapshot on the openAI website does not offer a date. To ensure consistency and reproducibility, we ran all experiments using this model again. Consequently, we updated all results across all prompts and revised our conclusions. Mainly, we observed that feasibility remains nearly perfect across all instance sizes. For other metrics, which are less obvious, there is a decrease, but less abrupt than before.  While overall performance has improved, 20 students is still an extremely small scale for matching problems, yet already demonstrates  increasing complexity. Thus, our overall conclusions remain valid. We still adapted the Section 4 under RQ1 to include the new observed results, and also added how base models don’t satisfy the more complex criteria, such as stability and optimality, under any number of students, showing they can’t reason over those type of preferences and constraints, but reasoning models do show great performance at small scale, although their performance becomes challenging very quickly. We therefore soften our conclusions, but the results still remain relevant as such problems are never at small scale, and although we don’t suspect that LLM will be used when there is already the DA algorithm, we raised concerns about using those as been done before in combinatorial optimization for heuristic generation in more complex settings.
>
> For the “General” prompt, results indicate that removing explicit algorithmic guidance does not generally improve performance and results are better under the basic prompt than the general one, with the exception of Qwen-QwQ. Table 4 reports performance across all models and prompts and reflects these findings. These updates are also discussed in Section 4 under RQ2.

---

> ### Author Response · Authors · 2026-04-11
> **Additional Prompt and Confidence Interval (3/3)**
>
> 3. Statistical Significance Analysis
>
> For each instance, we evaluate whether four metrics are satisfied, resulting in binary outcomes. We then report the proportion of metrics that meet these criteria. While binary outcomes typically require large sample sizes for statistical significance, we ensure that our analysis is never conducted at a small scale.
>
> Because we analyze multiple dimensions, such as the number of students, the ratio with the number of colleges, and preference lengths, sample sizes remain substantial within each category. For example, when analyzing the number of students, we have 72, 99, 99, and 99 samples for 5, 10, 15, and 20 students, respectively (with some configurations excluded for 5 students due to feasibility constraints), for each model and prompt combination. For preference length analysis, each category includes 123 samples, and the full dataset consists of 369 instances. When aggregating results across models or prompts, sample sizes increase further. This supports the statistical reliability of our conclusions.
>
> That said, reporting confidence measures remains important. To improve clarity, we updated Figures 5 and 6 by adding error bars representing 95% confidence intervals (CIs), computed using the Wilson method for proportions. For Figure 7, where lines are closely clustered, we added shaded regions to represent CIs. Additionally, we included Section D.3 in the appendix, which provides tables corresponding to each figure along with their 95% CIs. Specifically, Tables 5 and 6 report CIs for each prompt, aggregated either by model type (base vs. reasoning) or by individual model. Table 7 presents CIs across models and preference types, while Table 8 shows results by number of students and model.

---

### Review · Reviewer_kFHZ · 2026-05-12

**Summary Of Contributions:**

This paper introduces a benchmark of 369 instances for the College Admission Problem (CAP), a canonical many-to-one matching problem, and uses it to evaluate the reasoning capabilities of several open-weight LLMs. The benchmark varies four instance parameters: number of students, student-college ratio, student preference type, and college capacity regime, and evaluates models across nine prompting strategies, including Chain-of-Thought (CoT), In-Context Learning (ICL), and role-based prompting. Four hierarchical performance metrics are proposed: feasibility, assignment stability, matching stability, and student optimality. The authors find that reasoning-specialized (i.e. models with explicit known RL post-training) LLMs (QwQ 32B, GPT-oss 120B) substantially outperform base models (Llama-8B, 70B, Mistral 7B, Qwen 7B), all models degrade as instance size grows, and iterative prompting yields non-monotonic improvement.

**Audience:**

Yes

**Audience Explanation:**

Existing combinatorial optimization benchmarks focus almost exclusively on cost-minimization NP-hard problems (for example, TSP). The CAP is solvable in polynomial time yet imposes non-trivial reasoning constraints involving stability, feasibility, and preference satisfaction simultaneously. This makes it a natural and underexplored stress-test for structured reasoning, distinct from arithmetic or graph benchmarks.

**Claims And Evidence:**

Yes

**Claims Explanation:**

Most of the claims are well-supported by the extensive empirical analysis presented in the paper. The following claims should be relaxed to match the rigor justified based on available evidence, or lack thereof.

1. _Iterative prompting improvements are non-monotonic._  This is stated as a general claim, but the results show that there are some caveats to it. Looking at the data in Tables 9 to 16, for base models, this is true. Qwen 7B under Role prompting, for instance, shows feasibility going from 0.2398 (no iterative prompting) to 0.4503 (Last) to 0.5556 (Best) - meaning the last attempt frequently is not the best. The gap between Last and Best is meaningful and confirms non-monotonicity. However, for QwQ 32B, across virtually all prompt strategies, the Last and Best attempt columns are identical. This means QwQ converges to and stays at its best answer, which is monotonic improvement, not non-monotonic. The paper acknowledges this in passing ("it is more challenging to evaluate the impact of self-correctness ... from the reasoning models") but the abstract and introduction present non-monotonicity as a general finding, which is misleading given that the model with the most impressive iterative performance (QwQ) is actually a counterexample to the claim.

2. _Iterative prompting improves through answer diversity rather than genuine self-correction._ This is presented as a hypothesis in Section 4, but the paper provides no empirical analysis to support it. The argument is that the gap between the best and last attempts shows that models are not reliably converging, which could indicate diversity rather than correction. But the same gap is equally consistent with models that self-correct but then over-correct or degrade, which is a genuinely different mechanism. To support the diversity hypothesis, one would need to show, for instance, that best attempts are distributed across iteration steps rather than skewed toward later ones, or that successive answers are structurally dissimilar. Neither analysis is present in this paper.

3. _Smaller Llama 8B yields statistically significantly better feasibility results than the larger Llama 70B._
The proposed explanation, that smaller models prioritize explicit metrics whereas larger models may underperform by attempting to satisfy the more demanding ones, is a reasonable hypothesis but is purely speculative. No ablation or attention analysis is provided to support this interpretation. The paper should present it more explicitly as a hypothesis than an explanation.

**Requested Changes:**

There are some typo errors that have to be fixed.
1. Table 11, QwQ 32B - no iterative prompting. Please check the correctness of feasibility metric here.
2. In Fig 17, please clearly state which models were used.
3. Table 17 contains what appears to be a data entry error: the Mistral 7B and Llama 70B rows under "No iterative prompting" are identical to each other, and the Llama 70B rows appear to have been duplicated to fill the Mistral rows. The authors should verify and correct this table.
4. The radar charts in Figure 7 overlay nine prompt strategies with thin, overlapping lines of similar colour. This makes it nearly impossible to distinguish individual strategies, particularly for the base models where most lines collapse near the origin. Consider replacing these with grouped bar charts or small-multiples, or at minimum providing a version with clearer visual encoding (e.g., highlighting the top-2 and bottom-2 strategies per model type).

---

> ### Author Response · Authors · 2026-05-19
> **Claims: Iterative Prompting and Model Size**
>
> We would like to thank the reviewer for their constructive feedback and detailed annotations. We have revised the paper accordingly to address the points raised, please see the changes in color purple.
>
> **Monotonicity of iterative prompting**
>
> We thank the reviewer for this observation. While there are indeed cases where the last and best-matching outputs (out of a maximum of five iterations) coincide, there are also multiple instances where a performance gap emerges, as correctly noted by the reviewer. This remains an interesting finding.
>
> When feedback and previous attempts are incorporated into the model, we expect performance to improve; however, our results show that this is not consistently the case. This does not imply that improvement is impossible, and we have clarified this point in both the abstract and Section 4 (RQ3). Additionally, due to the strong performance of the reasoning model QwQ on small instances, in many cases the student-optimal matching is generated before reaching the maximal number of iterations allowed. In such cases, the iterative loop is stopped and naturally, the outcome between the best and last matching is the same. This clarification, along with the explicit statement that we do not claim iterative prompting always causes a degradation of performances, has been incorporated into Section 4 (RQ3).
>
> **Multi answers generation vs improvement**
>
> We thank the reviewer for their great point on this matter. It is true that we did not analyze further the distribution of cases in which models improve their answers versus those in which they do not.
>
> We are currently investigating this aspect by analyzing, for each model and prompt, the proportion of satisfied metrics (out of four) across iterations. Preliminary results support the hypothesis of diversity rather than systematic self-degradation, as the number of satisfied metrics remains relatively stable across iterations. Under a self-degradation hypothesis caused by over-correction, we would instead expect a decline.To avoid bias in cases where a successful match is achieved before the maximum number of iterations, a score of 1.0 is propagated to subsequent iterations. We also report, for each iteration, the proportion of instances that reached that iteration.
> In the meantime, we have revised the paper such that Section 4 (RQ3) now adjusts our claims to the observed experimental results on iterative prompting, while Section 5 (Discussion) outlines our hypothesis and the potential directions enabling to support or disprove them.
>
>
> **Model size for feasibility metric**
>
> We agree with the reviewer that it should be clearly stated that we only propose a hypothesis to explain the observed results, and that we do not provide further evidence to substantiate it. Accordingly, we have revised Section 4 (RQ1) to make this explicit, clarifying that further analysis would be needed to support that conclusion.

---

> > ### Author Response · Authors · 2026-05-25
> > **Updated Figure**
> >
> > The figures referenced in our previous response have now been added to the appendix, along with explicit references to them in the discussion section. Even though we do not observe noticeable degradation from overcorrection, we mention in the discussion that future work could further investigate how feedback is incorporated and how response distributions evolve across iterations.

---

> ### Author Response · Authors · 2026-05-19
> **Typos and Figure**
>
> We are very grateful for the reviewer to have caught some typos. We will briefly mention how it was corrected.
>
> - Table 11: For QwQ-32B, the feasibility should have been 0.9825 instead of 9825.0
> - Figure 17: In this figure, we aggregated over all models, and we changed the caption to make that known.
> - Table 17: There was indeed a copy pasted issue in the lines of Llama 70B and Mistral under this table. We have run again all experiments to make sure that it was the results of Llama70B, and have updated with the correct one for Mistral, observing the ordinary trend such as a very high assignment stability value and very close to other prompts.
> - Figure 7: In this Figure, we wanted to display the difference in prompting for every metrics and models. We agree with the reviewer that the Figure is not easily readable, and we had already added to the Appendix D.3 the tables 5 and 6 to present those results in more detail.
> While we did consider bar plots, we opted for spider plots to emphasize trends rather than absolute values.  Note that  we normalized the results in order to make trends clear.
> As briefly discussed in the paper, certain metrics (e.g., feasibility) are much easier to satisfy than others (e.g, optimality) and both have therefore very different values.  Without normalization, some metrics would visually dominate, while others would appear effectively negligible, even though the takeaway from this figure is not the absolute values but the relative differences across prompts.
> The relevant signal lies in their relative differences across prompts rather than in their absolute magnitudes.
> To improve the readability of the figure, instead of normalizing, we clipped the values at different ranges for each metric, still showing the range to not confuse the readers that results are comparable across each axis. We believe that this visualization choice, together with the detailed tables provided in the appendix, offers an effective way to analyze the effect of prompting. However, we are open to revising the presentation to bar plots, or at least include them in the appendix, if the reviewers believe this would improve clarity.

---

### Decision · Action_Editor_Ltc5 · 2026-07-01

**Recommendation:** Accept with minor revision

**Additional Comments:**

Other than the standard changes to produce the camera ready, the revision has two key aspects:

1) Two reviewers identified the lack of comparison to closed source models as a major limitation in terms of justifying the generality of claims.  The authors proposed using GPT 5-mini or Grok 4.1 Fast, which at least one reviewer and I agree would suffice to address this concern.  So I strongly encourage the authors to  implement that.  However, this is intended to be a minor revision process, so if this proves more onerous or expensive than anticipated at a minimum an appendix should be included discussing how the benchmarking could be conducted, the technical caveats, and the source of the expense (which can draw on the discussion already generated on this issue).

2) In the course or revisions some reproducibility issues were discovered which appear to be due to package versions.  An appendix has been added to document all necessary details to reproduce the results, but for some details it relies on the planned release of the code.  As part of the camera ready preparation please release the code so that I can verify everything required is included, which, per a reviewer, requires at a minimum discussion of "framework, quantization, hardware, pinned HF commit".  Both for me and for future users the README should incude a clear discussion of this with either all relevant details or pointers to where in the code they can be found.

**Audience:**

Yes

**Audience Explanation:**

The paper develops a benchmarking approach for LLM performance on a classic economic task, matching.  The paper is likely of interest both to the community interested in LLM benchmarking as well as those interested in the applications of LLMs in economic domains.

**Claims And Evidence:**

Yes

**Claims Explanation:**

Through discussions with the reviewers a number of claims have been adjusted to make sure they are consistent with the results shown in the paper.  There were two lingering concerns addressed in my revision below.